# Efficient Sign-Based Optimization: Accelerating Convergence via Variance Reduction

**Wei Jiang[1], Sifan Yang[1,2], Wenhao Yang[1,2], Lijun Zhang[1,3,2,*]**

[1]National Key Laboratory for Novel Software Technology, Nanjing University, Nanjing, China
[2]School of Artificial Intelligence, Nanjing University, Nanjing, China
[3]Pazhou Laboratory (Huangpu), Guangzhou, China
{jiangw, yangsf, yangwh, zhanglj}@lamda.nju.edu.cn

## Abstract

Sign stochastic gradient descent (signSGD) is a communication-efficient method that transmits only the sign of stochastic gradients for parameter updating. Existing literature has demonstrated that signSGD can achieve a convergence rate of $\mathcal{O}(d^{1/2}T^{-1/4})$, where $d$ represents the dimension and $T$ is the iteration number. In this paper, we improve this convergence rate to $\mathcal{O}(d^{1/2}T^{-1/3})$ by introducing the Sign-based Stochastic Variance Reduction (SSVR) method, which employs variance reduction estimators to track gradients and leverages their signs to update. For finite-sum problems, our method can be further enhanced to achieve a convergence rate of $\mathcal{O}(m^{1/4}d^{1/2}T^{-1/2})$, where $m$ denotes the number of component functions. Furthermore, we investigate the heterogeneous majority vote in distributed settings and introduce two novel algorithms that attain improved convergence rates of $\mathcal{O}(d^{1/2}T^{-1/2} + dn^{-1/2})$ and $\mathcal{O}(d^{1/4}T^{-1/4})$ respectively, outperforming the previous results of $\mathcal{O}(dT^{-1/4} + dn^{-1/2})$ and $\mathcal{O}(d^{3/8}T^{-1/8})$, where $n$ represents the number of nodes. Numerical experiments across different tasks validate the effectiveness of our proposed methods.

## 1 Introduction

This paper investigates the stochastic optimization problem

$$\min_{\mathbf{x}\in\mathbb{R}^d} f(\mathbf{x}), \tag{1}$$

where $f : \mathbb{R}^d \mapsto \mathbb{R}$ is a smooth and non-convex function. We assume that only noisy estimations of the gradient $\nabla f(\mathbf{x})$ can be accessed, represented as $\nabla f(\mathbf{x};\xi)$, where $\xi$ is a random sample drawn from a stochastic oracle such that $\mathbb{E}[\nabla f(\mathbf{x};\xi)] = \nabla f(\mathbf{x})$.

The most well-known method for problem (1) is stochastic gradient descent (SGD), which performs $\mathbf{x}_{t+1} = \mathbf{x}_t - \eta\nabla f(\mathbf{x}_t;\xi_t)$ for each iteration, where $\xi_t$ is the sample used in the $t$-th iteration, and $\eta$ is the learning rate. It has been proved that the SGD method can obtain a convergence rate of $\mathcal{O}(T^{-1/4})$ [Ghadimi and Lan, 2013], where $T$ is the iteration number. Recently, sign stochastic gradient descent (signSGD) method [Seide et al., 2014, Bernstein et al., 2018] has become popular in the machine learning community, which uses the sign of the stochastic gradient to update, i.e.,

$$\mathbf{x}_{t+1} = \mathbf{x}_t - \eta\,\mathrm{Sign}(\nabla f(\mathbf{x}_t;\xi_t)).$$

This method can largely reduce the communication overhead in distributed environments, and prior research [Bernstein et al., 2018, 2019] has established that signSGD can achieve a convergence rate

---

*Lijun Zhang is the corresponding author.

38th Conference on Neural Information Processing Systems (NeurIPS 2024).

of $\mathcal{O}(d^{1/2}T^{-1/4})$ measured in terms of the $l_1$-norm. Since the $\mathcal{O}(T^{-1/4})$ rate is already optimal for SGD methods when measured in the $l_2$-norm [Arjevani et al., 2023], we can not further improve the dependence on $T$ for signSGD method, considering that $\|\mathbf{x}\|_2 \leq \|\mathbf{x}\|_1$ for any $\mathbf{x}$.[2] However, it is also known that variance reduction techniques can further enhance the convergence rate to $\mathcal{O}(T^{-1/3})$, under a slightly stronger assumption of average smoothness [Fang et al., 2018, Wang et al., 2019, Cutkosky and Orabona, 2019]. This leads to a natural question: *Can the convergence of sign-based methods be further improved by employing variance reduction techniques along with the average smoothness assumption?* We respond affirmatively by introducing the Sign-based Stochastic Variance Reduction (SSVR) method. By integrating variance reduction technique [Cutkosky and Orabona, 2019] with sign operations, we achieve an improved convergence rate of $\mathcal{O}(d^{1/2}T^{-1/3})$ measured in the $l_1$-norm, matching the optimal rates in terms of $T$ for stochastic variance reduction methods [Fang et al., 2018, Li et al., 2021, Arjevani et al., 2023].

Furthermore, we investigate a special case of problem (1), in which the objective function exhibits a finite-sum structure:

$$\min_{\mathbf{x}\in\mathbb{R}^d} f(\mathbf{x}) = \frac{1}{m}\sum_{i=1}^m f_i(\mathbf{x}), \tag{2}$$

where each $f_i(\cdot)$ is smooth and non-convex. This problem has been extensively studied in stochastic optimization [Zhang et al., 2013, Defazio et al., 2014, Fang et al., 2018], but is less explored with sign-based methods. Previous literature proposes signSVRG [Chzhen and Schechtman, 2023] method to deal with the finite-sum problem, which achieves a convergence rate of $\mathcal{O}(m^{1/2}d^{1/2}T^{-1/2})$. However, its dependence on $m$ is sub-optimal, failing to match the $\mathcal{O}(m^{1/4}T^{-1/2})$ lower bound [Fang et al., 2018, Li et al., 2021] for problem (2). To address this gap, we propose the SSVR-FS algorithm, which periodically computes the exact gradient [Zhang et al., 2013, Johnson and Zhang, 2013] and incorporates it into the variance reduction estimator. In this way, we can achieve an improved convergence rate of $\mathcal{O}(m^{1/4}d^{1/2}T^{-1/2})$ for finite-sum problems.

Finally, sign-based methods are especially favorable in distributed settings, where the parameter server aggregates gradient signs from each worker through majority vote [Bernstein et al., 2018], allowing 1-bit compression of communication in both directions. Existing literature [Bernstein et al., 2018, 2019] has proved that signSGD can obtain a convergence rate of $\mathcal{O}(d^{1/2}T^{-1/4})$ for majority vote in homogeneous settings, where the data across nodes is uniformly distributed or identical. For the more challenging heterogeneous setting, in which data distribution can vary significantly across nodes, existing methods can only achieve convergence rates of $\mathcal{O}(dT^{-1/4} + dn^{-1/2})$ [Sun et al., 2023] and $\mathcal{O}(d^{3/8}T^{-1/8})$[3] [Jin et al., 2023], where $n$ denotes the number of nodes. Note that the first rate indicates that the gradient does not converge to zero as $T$ approaches infinity, and the second one suffers from a high sample complexity. To address these limitations, we first introduce our basic SSVR-MV method, which employs variance reduction estimators to track gradients and replaces the sign operation in each worker as a stochastic unbiased sign operation. This practice ensures 1-bit compression and unbiased estimation at the same time, and the newly proposed method can obtain an improved convergence rate of $\mathcal{O}(d^{1/2}T^{-1/2} + dn^{-1/2})$. By further substituting the sign operation in the parameter server with another stochastic unbiased sign operation, our method can further achieve a convergence rate of $\mathcal{O}(d^{1/4}T^{-1/4})$, which converges to zero as $T$ increases.

In summary, compared with existing methods, this paper makes the following contributions:

- For stochastic non-convex functions, we develop a sign-based variance reduction algorithm to achieve an improved convergence rate of $\mathcal{O}(d^{1/2}T^{-1/3})$, surpassing the $\mathcal{O}(d^{1/2}T^{-1/4})$ rate for signSGD methods.
- For non-convex finite-sum optimization, we further improve the our proposed method to obtain an enhanced convergence rate of $\mathcal{O}(m^{1/4}d^{1/2}T^{-1/2})$, which is better than the $\mathcal{O}(m^{1/2}d^{1/2}T^{-1/2})$ convergence rate for SignSVRG method.

---

[2]Note that most lower bounds are measured in the $l_2$-norm, and there are no existing lower bounds for the $l_1$-norm to the best of our knowledge. However, since $\|\mathbf{x}\|_1 \approx \sqrt{d}\|\mathbf{x}\|_2$ for dense vector $\mathbf{x}$, the additional $\mathcal{O}(\sqrt{d})$ term is acceptable when considering the $l_1$-norm. This is also the case for our methods (measured in the $l_1$-norm) and the lower bounds for variance reduction methods (for $l_2$-norm).

[3]The original convergence rate is measured under the squared $l_2$-norm, and we convert it to the rate under the $l_2$-norm criterion for a fair comparison.

Table 1: Summary of results for sign-based algorithms. Here, stochastic indicates problem (1), finite-sum represents problem (2), $N$ is the number of stochastic gradient calls, and $m$ is the number of component functions. Note that some rates are measured under squared $l_1$- or $l_2$-norm, and we convert them to $l_1$- or $l_2$-norm for a fair comparison.

| Method | Setting | Measure | Convergence rate |
|---|---|---|---|
| signSGD [Bernstein et al., 2018] | stochastic | $l_1$-norm | $\mathcal{O}\left(\frac{d^{1/2}}{N^{1/4}}\right)$ |
| EF-signSGD [Karimireddy et al., 2019] | stochastic | $l_2$-norm | $\mathcal{O}\left(\frac{1}{N^{1/4}}\right)$ |
| signSGD-SIM [Sun et al., 2023] | stochastic | $l_1$-norm | $\mathcal{O}\left(\frac{d}{N^{1/4}}\right)$ |
| SignSVRG [Chzhen and Schechtman, 2023] | finite-sum | $l_1$-norm | $\mathcal{O}\left(\frac{d^{1/2}m^{1/2}}{N^{1/2}}\right)$ |
| SignRVR/SignRVM [Qin et al., 2023] | finite-sum | $l_1$-norm | $\mathcal{O}\left(\frac{d^{1/2}m^{1/2}}{N^{1/2}}\right)$ |
| **Theorem 1** | stochastic | $l_1$-norm | $\mathcal{O}\left(\frac{d^{1/2}}{N^{1/3}}\right)$ |
| **Theorem 2** | finite-sum | $l_1$-norm | $\mathcal{O}\left(\frac{d^{1/2}m^{1/4}}{N^{1/2}}\right)$ |

Table 2: Summary of results for sign-based algorithms under the majority vote setting, where $n$ is the number of workers. Some rates are measured under squared $l_1$- or $l_2$-norm, and we convert them to $l_1$- or $l_2$-norm for a fair comparison.

| Method | Heterogeneous | Measure | Convergence rate |
|---|---|---|---|
| signSGD [Bernstein et al., 2018] | ✗ | $l_1$-norm | $\mathcal{O}\left(\frac{d^{1/2}}{T^{1/4}}\right)$ |
| Signum [Bernstein et al., 2019] | ✗ | $l_1$-norm | $\mathcal{O}\left(\frac{d^{1/2}}{T^{1/4}}\right)$ |
| SSDM [Safaryan and Richtarik, 2021] | ✓ | $l_2$-norm | $\mathcal{O}\left(\frac{d^{1/2}}{T^{1/4}}\right)$ |
| Sto-signSGD [Jin et al., 2023] | ✓ | $l_2$-norm | $\mathcal{O}\left(\frac{d^{3/8}}{T^{1/8}}\right)$ |
| MV-sto-signSGD-SIM [Sun et al., 2023] | ✓ | $l_1$-norm | $\mathcal{O}\left(\frac{d}{T^{1/4}} + \frac{d}{n^{1/2}}\right)$ |
| **Theorem 3** | ✓ | $l_1$-norm | $\mathcal{O}\left(\frac{d^{1/2}}{T^{1/2}} + \frac{d}{n^{1/2}}\right)$ |
| **Theorem 4** | ✓ | $l_2$-norm | $\mathcal{O}\left(\frac{d^{1/4}}{T^{1/4}}\right)$ |

- We also investigate sign-based variance reduction methods with heterogeneous majority vote in distributed settings. The proposed algorithms can obtain the convergence rates of $\mathcal{O}(d^{1/2}T^{-1/2} + dn^{-1/2})$ and $\mathcal{O}(d^{1/4}T^{-1/4})$, which outperform the previous results of $\mathcal{O}(dT^{-1/4} + dn^{-1/2})$ and $\mathcal{O}(d^{3/8}T^{-1/8})$, respectively.

We compare our results with existing methods in Table 1 and Table 2, and validate the effectiveness of our method via numerical experiments in Section 5.

## 2 Related work

This section provides an overview of the existing literature on signSGD methods and stochastic variance reduction techniques.

### 2.1 SignSGD and its variants

The idea of only transmitting the sign information of the stochastic gradient traces back to the 1-bit SGD algorithm, introduced by Seide et al. [2014]. Despite the biased nature of the sign operation, Bernstein et al. [2018] demonstrated that signSGD achieves a convergence rate of $\mathcal{O}(d^{1/2}T^{-1/4})$ by using large batch sizes in each iteration. Despite the theoretical assurance, Karimireddy et al. [2019] highlighted that signSGD may not converge to the optimal solutions for convex functions and could suffer from poor generalization without large batches. To address these issues, they proposed the

EF-signSGD method, which integrates error feedback into signSGD to correct errors introduced by the sign operation. Instead of requiring unbiased stochastic gradients in previous literature, Safaryan and Richtarik [2021] assumed that the signs of the stochastic gradient are the same as those of true gradient with a probability greater than $1/2$. Under this assumption, they demonstrated that signSGD can obtain a similar convergence rate but does not require large batches anymore. Recently, Sun et al. [2023] proposed the signSGD-SIM method, which incorporates the momentum into the signSGD, achieving a convergence rate of $\mathcal{O}(dT^{-1/4})$ with constant batch sizes and an improved convergence of $\mathcal{O}(d^{3/2}T^{-2/7})$ with second-order smoothness.

To deal with the finite-sum problems, Chzhen and Schechtman [2023] developed SignSVRG algorithm, which combines SVRG [Johnson and Zhang, 2013] method with signSGD and achieves a convergence rate of $\mathcal{O}(d^{1/2}m^{1/2}T^{-1/2})$, where $m$ is the number of component functions. More recently, Qin et al. [2023] further investigate signSGD with random reshuffling, achieving a convergence rate of $\mathcal{O}\left(m^{-1/2}T^{-1/2}\log(mT) + \|\sigma\|_1\right)$, where $\sigma$ is the variance bound of stochastic gradients. By leveraging variance-reduced gradients and momentum updates, they further propose the SignRVR and SignRVM methods, both achieving the convergence rate of $\mathcal{O}\left(d^{1/2}m^{1/2}T^{-1/2}\right)$.

In distributed settings, sign-based methods with majority vote are also widely investigated. Bernstein et al. [2018, 2019] first indicated that signSGD and its momentum variant Signum can enable 1-bit compression of worker-server communication, obtaining the $\mathcal{O}(d^{1/2}T^{-1/4})$ convergence rates in the homogeneous environment. For the more challenging heterogeneous settings, SSDM method [Safaryan and Richtarik, 2021] attains the same $\mathcal{O}(d^{1/2}T^{-1/4})$ convergence rate, but the information sent back to the server is not a sign information anymore. To remedy this issue, Sto-signSGD algorithm [Jin et al., 2023] is proposed, equipped with a convergence rate of $\mathcal{O}(d^{3/4}T^{-1/4})$ measured in squared $l_2$-norm. More recently, Sun et al. [2023] introduced the MV-signSGD-SIM algorithm and demonstrated a convergence rate of $\mathcal{O}(dT^{-1/4} + dn^{-1/2})$, which could be further enhanced to $\mathcal{O}(d^{3/2}T^{-2/7} + dn^{-1/2})$ under second-order smoothness conditions, where $n$ denotes the number of nodes in the distributed system.

## 2.2 Stochastic variance reduction methods

Stochastic variance reduction methods have gained significant attention in the optimization community in recent years. Among the pioneering approaches, the stochastic average gradient (SAG) method [Roux et al., 2012] and Stochastic Dual Coordinate Ascent (SDCA) algorithm [Shalev-Shwartz and Zhang, 2013] utilize a memory of previous gradients to ensure variance reduction, achieving linear convergence for strongly convex functions. To circumvent the need for storing gradients, the stochastic variance reduced gradient (SVRG) [Zhang et al., 2013, Johnson and Zhang, 2013] recalculates the full gradient periodically to enhance the accuracy of gradient estimators, maintaining linear convergence for strongly convex functions. Inspired by SAG and SVRG, Defazio et al. [2014] introduced the SAGA algorithm, which not only provides superior convergence rates but also supports proximal regularization. Subsequently, the stochastic recursive gradient algorithm (SARAH) [Nguyen et al., 2017] employs a simple recursive approach to update gradient estimators, ensuring better convergence for smooth convex functions.

For non-convex optimization, inspired by the SVRG algorithm, many methods [Reddi et al., 2016, Lei et al., 2017, Zhou et al., 2020] employ variance reduction to design their algorithms and provide the corresponding convergence guarantees. More recent well-known advancements include the SPIDER [Fang et al., 2018] and SpiderBoost [Wang et al., 2019] methods, which improved the $\mathcal{O}(T^{-1/4})$ convergence rate of traditional SGD to $\mathcal{O}(T^{-1/3})$ under the average smoothness assumption. The convergence rate can be further improved to $\mathcal{O}(m^{1/4}T^{-1/2})$ for problems with a finite-sum structure, where $m$ represents the number of component functions. However, these methods typically require a huge batch size to ensure convergence. To avoid this limitation, the stochastic recursive momentum (STORM) method [Cutkosky and Orabona, 2019] introduces a momentum-based updating and an adaptive learning rate based on the stochastic gradients, achieving a convergence rate of $\tilde{\mathcal{O}}(T^{-1/3})$ without necessitating large batches. More recently, variance reduction techniques are widely employed in more complex problems to improve the existing convergence rates, such as compositional optimization [Wang et al., 2017a,b, Yuan et al., 2019, Jiang et al., 2022a, 2023], multi-level optimization [Chen et al., 2021, Zhang and Xiao, 2021, Jiang et al., 2022b, 2024b], adaptive algorithms [Kavis et al., 2022, Jiang et al., 2024a], and distributionally robust optimization [Yu et al., 2024].

---

**Algorithm 1** SSVR

1: **Input:** time step $T$, initial point $\mathbf{x}_1$
2: **for** time step $t = 1$ **to** $T$ **do**
3:     Draw a batch of samples $\{\xi_t^1, \cdots, \xi_t^{B_1}\}$
4:     Compute $\mathbf{v}_t = \frac{1}{B_1} \sum_{k=1}^{B_1} \nabla f(\mathbf{x}_t; \xi_t^k) + (1 - \beta) \left( \mathbf{v}_{t-1} - \frac{1}{B_1} \sum_{k=1}^{B_1} \nabla f(\mathbf{x}_{t-1}; \xi_t^k) \right)$
5:     Update the decision variable: $\mathbf{x}_{t+1} = \mathbf{x}_t - \eta \operatorname{Sign}(\mathbf{v}_t)$
6: **end for**
7: Select $\tau$ uniformly at random from $\{1, \ldots, T\}$
8: Return $\mathbf{x}_\tau$

---

## 3  The proposed methods

In this section, we present the proposed methods for the expectation case, i.e., problem (1), and the finite-sum structure, i.e., problem (2), respectively, along with corresponding theoretical guarantees.

### 3.1  Sign-based stochastic variance reduction

In this subsection, we introduce our Sign-based Stochastic Variance Reduction (SSVR) method for problem (1). One crucial step in stochastic optimization is to track the gradient of the objective function. Here, we use a variance reduction gradient estimator $\mathbf{v}_t$ to evaluate the overall gradient $\nabla f(\mathbf{x}_t)$. In the first iteration ($t = 1$), the estimator is defined as $\mathbf{v}_1 = \frac{1}{B_0} \sum_{k=1}^{B_0} \nabla f(\mathbf{x}_1; \xi_1^k)$, where $B_0$ is the batch size used in the first iteration. For subsequent iterations ($t \geq 2$), $\mathbf{v}_t$ is updated in the style of STORM [Cutkosky and Orabona, 2019], i.e.,

$$\mathbf{v}_t = \frac{1}{B_1} \sum_{k=1}^{B_1} \nabla f(\mathbf{x}_t; \xi_t^k) + (1 - \beta) \left( \mathbf{v}_{t-1} - \frac{1}{B_1} \sum_{k=1}^{B_1} \nabla f(\mathbf{x}_{t-1}; \xi_t^k) \right),$$

where $\beta$ represents the momentum parameter and $B_1$ is the batch size. This method ensures that the expectation of the estimation error $\mathbb{E}[\|\mathbf{v}_t - \nabla f(\mathbf{x}_t)\|^2]$ would be reduced gradually. After obtaining the gradient estimator $\mathbf{v}_t$, we update the decision variable using the sign of $\mathbf{v}_t$:

$$\mathbf{x}_{t+1} = \mathbf{x}_t - \eta \operatorname{Sign}(\mathbf{v}_t).$$

The whole algorithm is outlined in Algorithm 1. Next, we introduce the following assumptions for our SSVR method, which are standard and commonly adopted in the analysis of variance reduction methods and stochastic non-convex optimization [Fang et al., 2018, Wang et al., 2019, Cutkosky and Orabona, 2019, Li et al., 2021].

**Assumption 1** *(Average smoothness)*

$$\mathbb{E}_\xi \left[ \|\nabla f(\mathbf{x}; \xi) - \nabla f(\mathbf{y}; \xi)\|^2 \right] \leq L^2 \|\mathbf{x} - \mathbf{y}\|^2.$$

**Assumption 2** *(Bounded variance)*

$$\mathbb{E}_\xi \left[ \|\nabla f(\mathbf{x}; \xi) - \nabla f(\mathbf{x})\|^2 \right] \leq \sigma^2.$$

With the above assumptions, we can obtain the theoretical guarantee for our method as stated below.

**Theorem 1** *Under Assumptions 1 and 2, by setting $\beta = \mathcal{O}(\frac{1}{T^{2/3}})$, $\eta = \mathcal{O}(\frac{1}{d^{1/2}T^{2/3}})$, $B_0 = \mathcal{O}(T^{1/3})$, and $B_1 = \mathcal{O}(1)$, our SSVR method ensures:*

$$\mathbb{E}\left[\|\nabla f(\mathbf{x}_\tau)\|_1\right] \leq \mathcal{O}\left(\frac{d^{1/2}}{T^{1/3}}\right).$$

**Remark:** This convergence rate surpasses the $\mathcal{O}(d^{1/2}T^{-1/4})$ rate achieved by previous sign-based methods [Bernstein et al., 2018, 2019], and it also outperforms the $\mathcal{O}(d^{3/2}T^{-2/7})$ convergence rate under the second-order smoothness condition [Sun et al., 2023]. Specifically, to ensure that $\mathbb{E}[\|\nabla f(\mathbf{x}_\tau)\|_1] \leq \epsilon$, our method requires a sample complexity of $\mathcal{O}(d^{3/2}\epsilon^{-3})$, which is much better than the $\mathcal{O}(d^2\epsilon^{-4})$ and $\mathcal{O}(d^{21/4}\epsilon^{-7/2})$ complexities of previous approaches.

---

**Algorithm 2** SSVR for Finite-Sum (SSVR-FS)

---
1: **Input:** time step $T$, initial point $\mathbf{x}_1$
2: **for** time step $t = 1$ **to** $T$ **do**
3:     **if** $t \mod I == 0$ **then**
4:         Set $\tau = t$ and compute $\nabla f(\mathbf{x}_\tau) = \frac{1}{m} \sum_{i=1}^{m} \nabla f_i(\mathbf{x}_\tau)$
5:     **end if**
6:     Sample $i_t$ randomly from $\{1, 2, \cdots, m\}$
7:     Compute gradient estimator $\mathbf{v}_t$ according to equation (3)
8:     Update the decision variable: $\mathbf{x}_{t+1} = \mathbf{x}_t - \eta \text{Sign}(\mathbf{v}_t)$
9: **end for**
10: Select $\varphi$ uniformly at random from $\{1, \ldots, T\}$
11: Return $\mathbf{x}_\varphi$

---

## 3.2 Sign-based stochastic variance reduction for finite-sum structure

We now extend our SSVR method to deal with the finite-sum structure in problem (2). In this context, we introduce the following assumption for each component function, which is standard and widely adopted in existing literature [Fang et al., 2018, Wang et al., 2019, Li et al., 2021].

**Assumption 3** *(Smoothness) For each $i \in \{1, 2, \cdots, m\}$, the gradient functions satisfy:*
$$\|\nabla f_i(\mathbf{x}) - \nabla f_i(\mathbf{y})\| \leq L\|\mathbf{x} - \mathbf{y}\|.$$

To handle the finite-sum problems, we retain the core structure of our SSVR method while incorporating elements from the SVRG [Zhang et al., 2013, Johnson and Zhang, 2013] approach. Specifically, we compute a full batch gradient at the first step and every $I$ iteration, i.e.,

$$\nabla f(\mathbf{x}_\tau) = \frac{1}{m} \sum_{i=1}^{m} \nabla f_i(\mathbf{x}_\tau).$$

For other iterations, we randomly select an index $i_t$ from the set $\{1, 2, \cdots, m\}$ and construct a variance reduction gradient estimator $\mathbf{v}_t$ as follows:

$$\mathbf{v}_t = \underbrace{\nabla f_{i_t}(\mathbf{x}_t) + (1-\beta)(\mathbf{v}_{t-1} - \nabla f_{i_t}(\mathbf{x}_{t-1}))}_{\text{STORM estimator}} - \underbrace{\beta\left(\nabla f_{i_t}(\mathbf{x}_\tau) - \nabla f(\mathbf{x}_\tau)\right)}_{\text{error correction}}. \tag{3}$$

The first two terms of $\mathbf{v}_t$ align with the STORM estimator, and the last term measures the difference of past gradients between the selected component function $\nabla f_{i_t}(\mathbf{x}_\tau)$ and the overall objective $\nabla f(\mathbf{x}_\tau)$. Note that the STORM estimator employs the component gradient $\nabla f_{i_t}(\mathbf{x}_t)$ to track the overall gradient $\nabla f(\mathbf{x}_t)$, which leads to an estimation error due to the gap between the component function and the overall objective. This gap can be effectively mitigated by the error correction term we introduced in equation (3). With such a design, we can obtain a better gradient estimation of the overall gradient, and ensure that the estimation error $\mathbb{E}[\|\nabla f(\mathbf{x}_t) - \mathbf{v}_t\|^2]$ can be reduced gradually. After computing $\mathbf{v}_t$, we utilize its sign information to update the decision variable. The detailed procedure is outlined in Algorithm 2. Next, we present the theoretical convergence for this method.

**Theorem 2** *Under Assumption 3, by setting $\beta = \mathcal{O}(\frac{1}{m})$, $I = m$, and $\eta = \mathcal{O}(\frac{1}{m^{1/4}d^{1/2}T^{1/2}})$, our algorithm ensures:*

$$\mathbb{E}[\|\nabla F(\mathbf{x}_\varphi)\|_1] \leq \mathcal{O}\left(\frac{m^{1/4}d^{1/2}}{T^{1/2}}\right).$$

**Remark:** To ensure $\mathbb{E}[\|\nabla F(\mathbf{x}_\varphi)\|_1] \leq \epsilon$, the sample complexity is $\mathcal{O}(m + \frac{d\sqrt{m}}{\epsilon^2})$, which improves over the $\mathcal{O}(\frac{dm}{\epsilon^2})$ complexity of the previous SignSVRG method [Chzhen and Schechtman, 2023].

## 4 Sign-based stochastic variance reduction with majority vote

Sign-based methods are advantageous in distributed settings for their low communication overhead, as they can only transmit sign information between nodes via majority vote. This section explores sign-based stochastic methods with majority vote, a typical example of distributed learning extensively

---

**Algorithm 3** SSVR-MV

---

1: **Input:** time step $T$, initial point $\mathbf{x}_1$
2: **for** time step $t = 1$ **to** $T$ **do**
3:     **On** node $j \in \{1, 2, \cdots, n\}$:
4:         Draw sample $\xi_t^j$ and compute $\mathbf{v}_t^j = \nabla f_j(\mathbf{x}_t; \xi_t^j) + (1 - \beta)\left(\mathbf{v}_{t-1}^j - \nabla f_j(\mathbf{x}_{t-1}; \xi_t^j)\right)$
5:         *Option 1:* Send $\mathrm{S}_R(\mathbf{v}_t^j)$ to the parameter server, where $R = 4G$
6:         *Option 2:* Send $\mathrm{S}_G(\hat{\mathbf{v}}_t^j)$ to the parameter server, where $\hat{\mathbf{v}}_t^j = \Pi_G[\mathbf{v}_t^j]$
7:     **On** parameter server:
8:         *Option 1:* Send $\mathbf{v}_t = \mathrm{Sign}\left(\frac{1}{n}\sum_{j=1}^n \mathrm{S}_R\left(\mathbf{v}_t^j\right)\right)$ to all nodes
9:         *Option 2:* Send $\mathbf{v}_t = \mathrm{S}_1\left(\frac{1}{n}\sum_{j=1}^n \mathrm{S}_G\left(\hat{\mathbf{v}}_t^j\right)\right)$ to all nodes
10:    **On** node $j \in \{1, 2, \cdots, n\}$:
11:       Update the decision variable $\mathbf{x}_{t+1} = \mathbf{x}_t - \eta\mathbf{v}_t$
12: **end for**
13: Select $\tau$ uniformly at random from $\{1, \ldots, T\}$
14: Return $\mathbf{x}_\tau$

---

studied in previous sign-based algorithms [Bernstein et al., 2018, 2019, Safaryan and Richtarik, 2021, Sun et al., 2023]. To begin with, we investigate the following distributed learning task:

$$\min_{\mathbf{x} \in \mathbb{R}^d} f(\mathbf{x}) \coloneqq \frac{1}{n}\sum_{j=1}^n f_j(\mathbf{x}), \quad f_j(\mathbf{x}) = \mathbb{E}_{\xi^j \sim \mathcal{D}_j}\left[f_j(\mathbf{x}; \xi^j)\right], \tag{4}$$

where $\mathcal{D}_j$ represents the data distribution for node $j$, and $f_j(\mathbf{x})$ is the corresponding loss function. Some previous studies [Bernstein et al., 2018, 2019] investigate the homogeneous setting, which assumes the data across each node is uniformly distributed or identical, ensuring that $\mathbb{E}[f_i(\mathbf{x})] = f(\mathbf{x})$. In contrast, this paper considers the more challenging heterogeneous setting [Jin et al., 2023, Sun et al., 2023], where data distributions can vary significantly across nodes.

For sign-based methods in distributed settings, each node $j$ computes a gradient estimator $\mathbf{v}_t^j$ and transmits its sign, i.e., $\mathrm{Sign}(\mathbf{v}_t^j)$, to the parameter server. Note that the server can not directly send the aggregate information $\sum_{j=1}^n \mathrm{Sign}(\mathbf{v}_t^j)$ back to each node, since it loses binary characteristic after summation. A natural solution is to apply another sign operation to update the decision variable as:

$$\mathbf{x}_{t+1} = \mathbf{x}_t - \eta\,\mathrm{Sign}\left(\frac{1}{n}\sum_{j=1}^n \mathrm{Sign}(\mathbf{v}_t^j)\right).$$

This process is called majority vote [Bernstein et al., 2018], as each worker votes on the sign of the gradient, with the server tallying these votes and broadcasting the decision back to the nodes. However, the sign operation introduces bias in the estimation, and employing it twice can significantly amplify this bias, particularly in a heterogeneous environment. Previous analysis [Chen et al., 2020] indicates that signSGD fails to converge in the heterogeneous setting. To deal with this problem, we introduce an unbiased sign operation $\mathrm{S}_R(\cdot)$, which is defined below.

**Definition 1** *For any vector $\mathbf{v}$ with $\|\mathbf{v}\|_\infty \leq R$, define the function mapping $\mathrm{S}_R(\mathbf{v})$ as:*

$$[\mathrm{S}_R(\mathbf{v})]_k = \begin{cases} +1, & \text{with probability } \frac{1}{2} + \frac{[\mathbf{v}]_k}{2R}, \\[2mm] -1, & \text{with probability } \frac{1}{2} - \frac{[\mathbf{v}]_k}{2R}. \end{cases} \tag{5}$$

**Remark:** This operation provides an unbiased estimation of $\mathbf{v}/R$, such that $\mathbb{E}[\mathrm{S}_R(\mathbf{v})] = \mathbf{v}/R$. It is worth noting that the function mapping is valid when $\|\mathbf{v}\|_\infty \leq R$, since the probability should always fall within $[0, 1]$. For this purpose, we need to further assume that the gradient is bounded.

Utilizing this unbiased sign operation, we can update the decision variable as:

$$\mathbf{x}_{t+1} = \mathbf{x}_t - \eta \, \mathrm{Sign}\left(\frac{1}{n}\sum_{j=1}^{n} \mathrm{S}_R(\mathbf{v}_t^j)\right).$$

After applying $\mathrm{S}_R(\cdot)$, the output is a sign information, which can be transported between nodes efficiently. The complete algorithm, named SSVR with majority vote (SSVR-MV), is described in Algorithm 3 (with *Option 1*). Note that in Step 4, we set $\mathbf{v}_1^j = \nabla f(\mathbf{x}_1; \xi_1^j)$ when $t = 1$. Next, we present the convergence guarantee for the proposed algorithm with the following assumption.

**Assumption 4** *For each node $j$, the stochastic gradient is bounded by $G$ in the infinity norm, such that $\|\nabla f_j(\mathbf{x}; \xi)\|_\infty \leq G$.*

**Theorem 3** *Under Assumptions 1, 2 and 4, by setting $\beta = \frac{1}{2}$ and $\eta = \mathcal{O}(\frac{1}{T^{1/2}d^{1/2}})$, our SSVR-MV method (with Option 1) ensures:*

$$\mathbb{E}\left[\|\nabla f(\mathbf{x}_\tau)\|_1\right] \leq \mathcal{O}\left(\frac{d^{1/2}}{T^{1/2}} + \frac{d}{n^{1/2}}\right).$$

**Remark:** Our rate is better than the previous result of $\mathcal{O}(dT^{-1/4} + dn^{-1/2})$, and also outperforms the rate of $\mathcal{O}(d^{3/2}T^{-2/7} + dn^{-1/2})$ under the second-order smoothness [Sun et al., 2023].

Although the above convergence rate is superior to previous results, we have to note that the gradient does not converge to zero even as $T \to \infty$. To address this issue, we propose replacing another sign operation with the $\mathrm{S}_1(\cdot)$ mapping, as defined in equation (5) with $R = 1$. Additionally, in our prior analysis, we ensured that each $\mathbf{v}_t^j$ is bounded by assuming the stochastic gradient is bounded and using a constant $\beta$. Here, we instead suppose that the true gradient is bounded, as detailed below.

**Assumption 4′** *For each node $j$, the gradient is bounded such that $\|\nabla f_j(\mathbf{x})\| \leq G$.*

**Remark:** This assumption is weaker than the one used by Sun et al. [2023], which assumes all *stochastic* gradients are bounded, i.e., $\|\nabla f_j(\mathbf{x}; \xi)\| \leq G$.

To ensure each gradient estimator is bounded, we employ a projection operation $\hat{\mathbf{v}}_t^j = \Pi_G[\mathbf{v}_t^j]$, where $\Pi_G$ denotes the projection onto a ball of radius $G$. This allows us to utilize an unbiased sign mapping $\mathrm{S}_G(\hat{\mathbf{v}}_t^j)$ before transmission to the parameter server. The revised algorithm is presented in Algorithm 3 (with *Option 2*), and the modifications lie in Steps 6 and 9. We now present the convergence guarantee for this modified approach below.

**Theorem 4** *Under Assumptions 1, 2 and 4′, by setting $\beta = \mathcal{O}(\frac{1}{T^{1/2}})$ and $\eta = \mathcal{O}(\frac{1}{d^{1/2}T^{1/2}})$, our SSVR-MV method (with Option 2) ensures:*

$$\mathbb{E}\left[\|\nabla f(\mathbf{x}_\tau)\|\right] \leq \mathcal{O}\left(\frac{d^{1/4}}{T^{1/4}}\right).$$

**Remark:** This rate converges to zero as $T \to \infty$, and offers a significant improvement over the previous results of $\mathcal{O}(d^{3/8}T^{-1/8})$ [Jin et al., 2023]. Our result is also better than the $\mathcal{O}(d^{1/2}T^{-1/4})$ convergence rate obtained by Safaryan and Richtarik [2021], whose algorithm requires transmitting $\sum_{j=1}^{n} \mathrm{sign}(\mathbf{v}_t^j)$ back to all nodes, which is actually not sign information anymore.

## 5   Experiments

In this section, we assess the performance of the proposed methods through numerical experiments. We first evaluate the SSVR and SSVR-FS algorithms within the centralized setting, and then assess the performance of SSVR-MV method in the distributed learning environment. All experiments are conducted on NVIDIA 3090 GPUs.

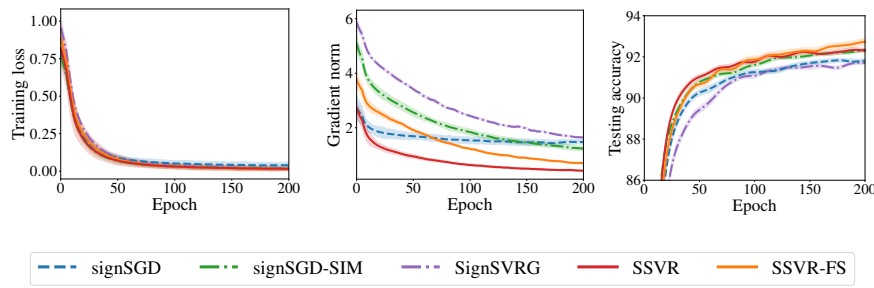

Figure 1: Results for CIFAR-10 dataset in the centralized environment.

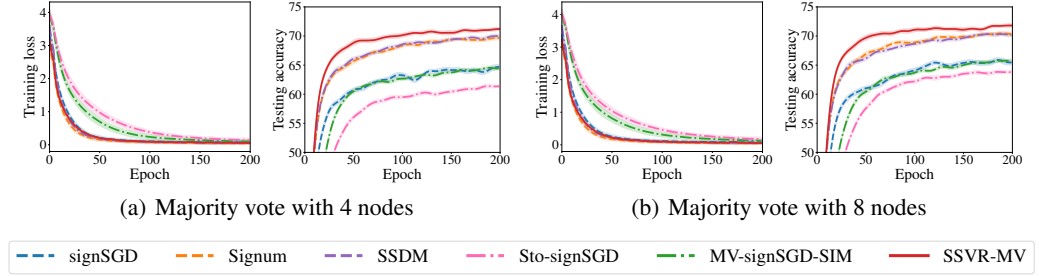

(a) Majority vote with 4 nodes       (b) Majority vote with 8 nodes

Figure 2: Results for CIFAR-100 dataset in the distributed environment.

## 5.1 Evaluation of SSVR and SSVR-FS methods in the centralized environment

To begin with, we conduct numerical experiments on multi-class image classification tasks to validate the effectiveness of our proposed methods. Concretely, we train a ResNet18 model [He et al., 2016] on the CIFAR-10 dataset [Krizhevsky, 2009]. We compare the performance of our SSVR and SSVR-FS methods against signSGD [Bernstein et al., 2018], signSGD-SIM [Sun et al., 2023], and SignSVRG [Chzhen and Schechtman, 2023]. For hyper-parameter tuning, we either follow the recommendations from the original papers or employ a grid search to determine the best settings. Specifically, the momentum parameter $\beta$ is searched from the set $\{0.1, 0.5, 0.9, 0.99\}$, and the learning rate is fine-tuned within the range of $\{1e-5, 1e-4, 1e-3, 1e-2, 1e-1\}$.

**Results.** The training loss, gradient norm, and testing accuracy are presented in Figure 1, with curves averaged over five runs. We observe that all methods exhibit a rapid decrease in training losses, with our methods showing a more pronounced reduction in the gradient norm. In terms of testing accuracy, our SSVR algorithm outperforms other sign-based methods, and our SSVR-FS method achieves superior accuracy in the final epochs.

## 5.2 Evaluation of SSVR-MV method in the distributed learning

Subsequently, we conduct experiments to evaluate the effectiveness of the SSVR-MV method in the distributed environment. Specifically, we train a ResNet50 model [He et al., 2016] on the CIFAR-100 dataset [Krizhevsky, 2009] with 4 and 8 nodes respectively. We compare the performance of our method against signSGD (with majority vote) [Bernstein et al., 2018], Signum (with majority vote) [Bernstein et al., 2019], SSDM [Safaryan and Richtarik, 2021], Sto-signSGD [Jin et al., 2023], and MV-signSGD-SIM [Sun et al., 2023]. The hyper-parameter tuning follows the same methodology as in the centralized environment experiment.

**Results.** We plot the training loss and testing accuracy in Figure 2, with all curves averaged over five runs. The results indicate that the training loss of our SSVR-MV algorithm decreases rapidly, and our method obtains higher testing accuracy compared to other methods, both in experiments with 4 nodes and 8 nodes.

# 6   Conclusion

In this paper, we explore sign-based stochastic variance reduction (SSVR) methods, which use only the sign information of variance-reduced estimators to update decision variables. The proposed method achieves an improved convergence rate of $\mathcal{O}(d^{1/2}T^{-1/3})$, surpassing the $\mathcal{O}(d^{1/2}T^{-1/4})$ convergence rate of signSGD methods. When applied to finite-sum problems, this rate can be further enhanced to $\mathcal{O}(m^{1/4}d^{1/2}T^{-1/2})$, which is also better than the $\mathcal{O}(m^{1/2}d^{1/2}T^{-1/2})$ convergence rate of SignSVRG. Finally, we investigate the SSVR method in distributed settings and devise novel algorithms to attain convergence rates of $\mathcal{O}(d^{1/2}T^{-1/2} + dn^{-1/2})$ and $\mathcal{O}(d^{1/4}T^{-1/4})$, which improve upon the previous results of $\mathcal{O}(dT^{-1/4} + dn^{-1/2})$ and $\mathcal{O}(d^{3/8}T^{-1/8})$ respectively.

## Acknowledgements

This work was partially supported by NSFC (62122037), the Collaborative Innovation Center of Novel Software Technology and Industrialization, and the Postgraduate Research & Practice Innovation Program of Jiangsu Province (No. KYCX24_0231).

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

# A    Proof of Theorem 1

Firstly, note that Assumption 1 indicates that the objective function $f(\mathbf{x})$ is also $L$-smooth [Li et al., 2021]. Given this property, we have that

$$
\begin{aligned}
&f(\mathbf{x}_{t+1}) \\
&\leq f(\mathbf{x}_t) + \langle \nabla f(\mathbf{x}_t), \mathbf{x}_{t+1} - \mathbf{x}_t \rangle + \frac{L}{2} \| \mathbf{x}_{t+1} - \mathbf{x}_t \|^2 \\
&\leq f(\mathbf{x}_t) + \langle \nabla f(\mathbf{x}_t), -\eta \operatorname{Sign}(\mathbf{v}_t) \rangle + \frac{\eta^2 L}{2} \| \operatorname{Sign}(\mathbf{v}_t) \|^2 \\
&\leq f(\mathbf{x}_t) + \eta \langle \nabla f(\mathbf{x}_t), \operatorname{Sign}(\nabla f(\mathbf{x}_t)) - \operatorname{Sign}(\mathbf{v}_t) \rangle - \eta \langle \nabla f(\mathbf{x}_t), \operatorname{Sign}(\nabla f(\mathbf{x}_t)) \rangle + \frac{\eta^2 Ld}{2} \quad (6) \\
&= f(\mathbf{x}_t) + \eta \langle \nabla f(\mathbf{x}_t), \operatorname{Sign}(\nabla f(\mathbf{x}_t)) - \operatorname{Sign}(\mathbf{v}_t) \rangle - \eta \| \nabla f(\mathbf{x}_t) \|_1 + \frac{\eta^2 Ld}{2} \\
&\leq f(\mathbf{x}_t) + 2\eta\sqrt{d} \| \nabla f(\mathbf{x}_t) - \mathbf{v}_t \| - \eta \| \nabla f(\mathbf{x}_t) \|_1 + \frac{\eta^2 Ld}{2},
\end{aligned}
$$

where the last inequality is due to the fact that

$$
\begin{aligned}
&\langle \nabla f(\mathbf{x}_t), \operatorname{Sign}(\nabla f(\mathbf{x}_t)) - \operatorname{Sign}(\mathbf{v}_t) \rangle \\
&= \sum_{i=1}^{d} \langle [\nabla f(\mathbf{x}_t)]_i, \operatorname{Sign}([\nabla f(\mathbf{x}_t)]_i) - \operatorname{Sign}([\mathbf{v}_t]_i) \rangle \\
&\leq \sum_{i=1}^{d} 2 \left| [\nabla f(\mathbf{x}_t)]_i \right| \cdot \mathbb{I} \left( \operatorname{Sign}([\nabla f(\mathbf{x}_t)]_i) \neq \operatorname{Sign}([\mathbf{v}_t]_i) \right) \\
&\leq \sum_{i=1}^{d} 2 \left| [\nabla f(\mathbf{x}_t)]_i - [\mathbf{v}_t]_i \right| \cdot \mathbb{I} \left( \operatorname{Sign}([\nabla f(\mathbf{x}_t)]_i) \neq \operatorname{Sign}([\mathbf{v}_t]_i) \right) \quad (7) \\
&\leq \sum_{i=1}^{d} 2 \left| [\nabla f(\mathbf{x}_t)]_i - [\mathbf{v}_t]_i \right| \\
&= 2 \| \nabla f(\mathbf{x}_t) - \mathbf{v}_t \|_1 \\
&\leq 2\sqrt{d} \| \nabla f(\mathbf{x}_t) - \mathbf{v}_t \|.
\end{aligned}
$$

Summing up and rearranging the equation (6), we derive:

$$
\begin{aligned}
&\mathbb{E} \left[ \frac{1}{T} \sum_{t=1}^{T} \| \nabla f(\mathbf{x}_t) \|_1 \right] \\
&\leq \frac{f(\mathbf{x}_1) - f(\mathbf{x}_{T+1})}{\eta T} + 2\sqrt{d} \cdot \mathbb{E} \left[ \frac{1}{T} \sum_{t=1}^{T} \| \nabla f(\mathbf{x}_t) - \mathbf{v}_t \| \right] + \frac{\eta Ld}{2} \quad (8) \\
&\leq \frac{\Delta_f}{\eta T} + 2\sqrt{d} \cdot \sqrt{\mathbb{E} \left[ \frac{1}{T} \sum_{t=1}^{T} \| \nabla f(\mathbf{x}_t) - \mathbf{v}_t \|^2 \right]} + \frac{\eta Ld}{2}
\end{aligned}
$$

where we define $\Delta_f = f(\mathbf{x}_1) - f_*$, and the second inequality is due to Jensen's Inequality.

Next, we can bound the term $\mathbb{E}\left[\frac{1}{T}\sum_{t=1}^{T}\|\nabla f(\mathbf{x}_t) - \mathbf{v}_t\|^2\right]$ as follows.

$$\mathbb{E}\left[\|\nabla f(\mathbf{x}_{t+1}) - \mathbf{v}_{t+1}\|^2\right]$$

$$= \mathbb{E}\left[\left\|(1-\beta)\mathbf{v}_t + \frac{1}{B_1}\sum_{k=1}^{B_1}\nabla f(\mathbf{x}_{t+1};\xi_{t+1}^k) - (1-\beta)\frac{1}{B_1}\sum_{k=1}^{B_1}\nabla f(\mathbf{x}_t;\xi_{t+1}^k) - \nabla f(\mathbf{x}_{t+1})\right\|^2\right]$$

$$= \mathbb{E}\left[\left\|(1-\beta)(\mathbf{v}_t - \nabla f(\mathbf{x}_t)) + \beta\frac{1}{B_1}\sum_{k=1}^{B_1}\left(\nabla f(\mathbf{x}_t;\xi_{t+1}^k) - \nabla f(\mathbf{x}_t)\right)\right.\right.$$

$$\left.\left. + \left(\nabla f(\mathbf{x}_t) - \nabla f(\mathbf{x}_{t+1}) + \frac{1}{B_1}\sum_{k=1}^{B_1}\nabla f(\mathbf{x}_{t+1};\xi_{t+1}^k) - \frac{1}{B_1}\sum_{k=1}^{B_1}\nabla f(\mathbf{x}_t;\xi_{t+1}^k)\right)\right\|^2\right]$$

$$\leq (1-\beta)^2\mathbb{E}\left[\|\mathbf{v}_t - \nabla f(\mathbf{x}_t)\|^2\right] + 2\beta^2\mathbb{E}\left[\left\|\frac{1}{B_1}\sum_{k=1}^{B_1}\nabla f(\mathbf{x}_t;\xi_{t+1}^k) - \nabla f(\mathbf{x}_t)\right\|^2\right]$$

$$+ 2\mathbb{E}\left[\left\|\frac{1}{B_1}\sum_{k=1}^{B_1}\left(\nabla f(\mathbf{x}_{t+1};\xi_{t+1}^k) - \nabla f(\mathbf{x}_t;\xi_{t+1}^k)\right)\right\|^2\right]$$

$$\leq (1-\beta)\mathbb{E}\left[\|\mathbf{v}_t - \nabla f(\mathbf{x}_t)\|^2\right] + \frac{2\beta^2\sigma^2}{B_1} + \frac{2L^2\|\mathbf{x}_{t+1} - \mathbf{x}_t\|^2}{B_1}$$

$$\leq (1-\beta)\mathbb{E}\left[\|\mathbf{v}_t - \nabla f(\mathbf{x}_t)\|^2\right] + \frac{2\beta^2\sigma^2}{B_1} + \frac{2L^2\eta^2 d}{B_1},$$

where the first inequality is due to the fact $\mathbb{E}\left[\left(\beta\frac{1}{B_1}\sum_{k=1}^{B_1}\left(\nabla f(\mathbf{x}_t;\xi_{t+1}^k) - \nabla f(\mathbf{x}_t)\right) + \nabla f(\mathbf{x}_t)\right.\right.$ $\left.\left.-\nabla f(\mathbf{x}_{t+1}) + \frac{1}{B_1}\sum_{k=1}^{B_1}\left(\nabla f(\mathbf{x}_{t+1};\xi_{t+1}^k) - \nabla f(\mathbf{x}_t;\xi_{t+1}^k)\right)\right)\right] = 0$, and $(a+b)^2 \leq 2a^2 + 2b^2$.

Summing up and noticing that we use a batch size of $B_0$ in the first iteration, we can ensure

$$\mathbb{E}\left[\frac{1}{T}\sum_{t=1}^{T}\|\mathbf{v}_t - \nabla f(\mathbf{x}_t)\|^2\right] \leq \frac{\mathbb{E}\left[\|\mathbf{v}_1 - \nabla f(\mathbf{x}_1)\|^2\right]}{\beta T} + \frac{2\sigma^2\beta}{B_1} + \frac{2L^2\eta^2 d}{\beta B_1} \tag{9}$$

$$\leq \frac{\sigma^2}{B_0\beta T} + \frac{2\sigma^2\beta}{B_1} + \frac{2L^2\eta^2 d}{\beta B_1}$$

Incorporating the above into equation (8) and setting that $\beta = \mathcal{O}\left(T^{-2/3}\right)$, $\eta = \mathcal{O}\left(d^{-1/2}T^{-2/3}\right)$, $B_0 = \mathcal{O}\left(T^{1/3}\right)$, $B_1 = \mathcal{O}\left(1\right)$, we observe:

$$\mathbb{E}\left[\frac{1}{T}\sum_{t=1}^{T}\|\nabla f(\mathbf{x}_t)\|_1\right] \leq \frac{\Delta_f}{\eta T} + 2\sqrt{d}\cdot\sqrt{\mathbb{E}\left[\frac{1}{T}\sum_{t=1}^{T}\|\nabla f(\mathbf{x}_t) - \mathbf{v}_t\|^2\right]} + \frac{\eta L d}{2}$$

$$\leq \frac{\Delta_f}{\eta T} + 2\sqrt{d}\cdot\sqrt{\frac{\sigma^2}{B_0\beta T} + \frac{2\sigma^2\beta}{B_1} + \frac{2L^2\eta^2 d}{\beta B_1}} + \frac{\eta L d}{2}$$

$$= \mathcal{O}\left(\frac{(\Delta_f + \sigma + L)d^{1/2}}{T^{1/3}}\right)$$

$$= \mathcal{O}\left(\frac{d^{1/2}}{T^{1/3}}\right),$$

which finishes the proof of Theorem 1.

## B Proof of Theorem 2

To improve the convergence rate for finite-sum structures, we can reuse the results of equation (8), but bound the term $\mathbb{E}\left[\frac{1}{T}\sum_{t=1}^{T}\|\nabla f(\mathbf{x}_t) - \mathbf{v}_t\|^2\right]$ differently. Since $\mathbf{v}_t = (1-\beta)\mathbf{v}_{t-1} + \beta\mathbf{h}_t + (1-$

$\beta) (\nabla f_{i_t}(\mathbf{x}_t) - \nabla f_{i_t}(\mathbf{x}_{t-1}))$, where $\mathbf{h}_t = \nabla f_{i_t}(\mathbf{x}_t) - \nabla f_{i_t}(\mathbf{x}_\tau) + \nabla f(\mathbf{x}_\tau)$, we have:

$$\mathbb{E}\left[\|\nabla f(\mathbf{x}_{t+1}) - \mathbf{v}_{t+1}\|^2\right]$$

$$= \mathbb{E}\left[\|(1-\beta)\mathbf{v}_t + \beta\mathbf{h}_{t+1} + (1-\beta)(\nabla f_{i_{t+1}}(\mathbf{x}_{t+1}) - \nabla f_{i_{t+1}}(\mathbf{x}_t)) - \nabla f(\mathbf{x}_{t+1})\|^2\right]$$

$$= \mathbb{E}\left[\|(1-\beta)(\mathbf{v}_t - \nabla f(\mathbf{x}_t)) + \beta\left(\mathbf{h}_{t+1} - \nabla f(\mathbf{x}_{t+1})\right)\right.$$
$$\left. + (1-\beta)\left(\nabla f(\mathbf{x}_t) - \nabla f(\mathbf{x}_{t+1}) + \nabla f_{i_{t+1}}(\mathbf{x}_{t+1}) - \nabla f_{i_{t+1}}(\mathbf{x}_t)\right)\|^2\right]$$

$$\leq (1-\beta)^2\mathbb{E}\left[\|\mathbf{v}_t - \nabla f(\mathbf{x}_t)\|^2\right] + 2\beta^2\mathbb{E}\left[\|\mathbf{h}_{t+1} - \nabla f(\mathbf{x}_{t+1})\|^2\right]$$
$$+ 2(1-\beta)^2\mathbb{E}\left[\|\nabla f(\mathbf{x}_t) - \nabla f(\mathbf{x}_{t+1}) + \nabla f_{i_{t+1}}(\mathbf{x}_{t+1}) - \nabla f_{i_{t+1}}(\mathbf{x}_t)\|^2\right]$$

$$\leq (1-\beta)^2\mathbb{E}\left[\|\mathbf{v}_t - \nabla f(\mathbf{x}_t)\|^2\right] + 2\beta^2\mathbb{E}\left[\|\mathbf{h}_{t+1} - \nabla f(\mathbf{x}_{t+1})\|^2\right]$$
$$+ 2(1-\beta)^2\mathbb{E}\left[\|\nabla f_{i_{t+1}}(\mathbf{x}_{t+1}) - \nabla f_{i_{t+1}}(\mathbf{x}_t)\|^2\right]$$

$$\leq (1-\beta)\mathbb{E}\left[\|\mathbf{v}_t - \nabla f(\mathbf{x}_t)\|^2\right] + 2\beta^2\mathbb{E}\left[\|\mathbf{h}_{t+1} - \nabla f(\mathbf{x}_{t+1})\|^2\right] + 2L^2\mathbb{E}\left[\|\mathbf{x}_{t+1} - \mathbf{x}_t\|^2\right]$$

$$\leq (1-\beta)\mathbb{E}\left[\|\mathbf{v}_t - \nabla f(\mathbf{x}_t)\|^2\right] + 2\beta^2 L^2\mathbb{E}\left[\|\mathbf{x}_{t+1} - \mathbf{x}_\tau\|^2\right] + 2L^2\mathbb{E}\left[\|\mathbf{x}_{t+1} - \mathbf{x}_t\|^2\right],$$

where the last inequality is due to the fact that:

$$\mathbb{E}\left[\|\mathbf{h}_{t+1} - \nabla f(\mathbf{x}_{t+1})\|^2\right] = \mathbb{E}\left[\|\nabla f_{i_{t+1}}(\mathbf{x}_{t+1}) - \nabla f_{i_{t+1}}(\mathbf{x}_\tau) + \nabla f(\mathbf{x}_\tau) - \nabla f(\mathbf{x}_{t+1})\|^2\right]$$
$$\leq \mathbb{E}\left[\|\nabla f_{i_{t+1}}(\mathbf{x}_{t+1}) - \nabla f_{i_{t+1}}(\mathbf{x}_\tau)\|^2\right]$$
$$\leq L^2\mathbb{E}\left[\|\mathbf{x}_{t+1} - \mathbf{x}_\tau\|^2\right].$$

By rearranging and summing up, we establish:

$$\mathbb{E}\left[\frac{1}{T}\sum_{t=1}^{T}\|\nabla f(\mathbf{x}_t) - \mathbf{v}_t\|^2\right]$$

$$\leq \frac{\mathbb{E}\left[\|\mathbf{v}_1 - \nabla f(\mathbf{x}_1)\|^2\right]}{\beta T} + 2\beta L^2\mathbb{E}\left[\frac{1}{T}\sum_{t=1}^{T}\|\mathbf{x}_{t+1} - \mathbf{x}_\tau\|^2\right] + \frac{2L^2}{\beta}\mathbb{E}\left[\frac{1}{T}\sum_{t=1}^{T}\|\mathbf{x}_{t+1} - \mathbf{x}_t\|^2\right]$$

$$\leq 2\beta I^2 L^2\mathbb{E}\left[\frac{1}{T}\sum_{t=1}^{T}\|\mathbf{x}_{t+1} - \mathbf{x}_t\|^2\right] + \frac{2L^2}{\beta}\mathbb{E}\left[\frac{1}{T}\sum_{t=1}^{T}\|\mathbf{x}_{t+1} - \mathbf{x}_t\|^2\right]$$

$$\leq 2L^2\left(\beta I^2 + \frac{1}{\beta}\right)\eta^2 d,$$

where we use full batch in the first iteration, and the second inequality is due to the fact that

$$\frac{1}{T}\sum_{t=1}^{T}\|\mathbf{x}_{t+1} - \mathbf{x}_\tau\|^2 = \frac{1}{T}\sum_{t=1}^{T}\left\|\sum_{i=\tau}^{t}(\mathbf{x}_{i+1} - \mathbf{x}_i)\right\|^2 \leq \frac{1}{T}\sum_{t=1}^{T}I\sum_{i=\tau}^{t}\|\mathbf{x}_{i+1} - \mathbf{x}_i\|^2$$

$$\leq \frac{I^2}{T}\sum_{t=1}^{T}\|\mathbf{x}_{t+1} - \mathbf{x}_t\|^2.$$

Incorporate the above into equation (8) and setting $I = m$, $\beta = \mathcal{O}\left(\frac{1}{m}\right)$, and $\eta = \mathcal{O}\left(\frac{1}{m^{1/4}d^{1/2}T^{1/2}}\right)$, we refine the bound as:

$$\mathbb{E}\left[\frac{1}{T}\sum_{t=1}^{T}\|\nabla f(\mathbf{x}_t)\|_1\right] \leq \frac{\Delta_f}{\eta T} + 2\sqrt{d}\cdot\sqrt{\mathbb{E}\left[\frac{1}{T}\sum_{t=1}^{T}\|\nabla f(\mathbf{x}_t) - \mathbf{v}_t\|^2\right]} + \frac{\eta L d}{2}$$

$$\leq \frac{\Delta_f}{\eta T} + 2\sqrt{d}\cdot\sqrt{2L^2\left(\beta I^2 + \frac{1}{\beta}\right)\eta^2 d} + \frac{\eta L d}{2}$$

$$= \mathcal{O}\left(\frac{(\Delta_f + L)\, m^{1/4}d^{1/2}}{T^{1/2}}\right)$$

$$= \mathcal{O}\left(\frac{m^{1/4}d^{1/2}}{T^{1/2}}\right).$$

## C    Proof of Theorem 3

By setting $\beta = \frac{1}{2}$, we can ensure that $\left\|\mathbf{v}_t^j\right\|_\infty \le R = 4G$, since

$$
\begin{aligned}
\left\|\mathbf{v}_t^j\right\|_\infty &= \left\|(1-\beta)\mathbf{v}_{t-1}^j + \nabla f(\mathbf{x}_t; \xi_t^j) - (1-\beta)f(\mathbf{x}_{t-1}; \xi_t^j)\right\|_\infty \\
&\le (1-\beta)\left\|\mathbf{v}_{t-1}^j\right\|_\infty + (2-\beta)G \\
&\le (1-\beta)^{t-1}\left\|\mathbf{v}_1^j\right\|_\infty + (2-\beta)G\sum_{s=1}^{t}(1-\beta)^{t-s} \\
&\le G + \frac{(2-\beta)G}{\beta} \le 4G = R.
\end{aligned}
$$

Since the overall objective function $f(\mathbf{x})$ is $L$-smooth, we have the following:

$$
\begin{aligned}
f(\mathbf{x}_{t+1}) \le &f(\mathbf{x}_t) + \langle \nabla f(\mathbf{x}_t), \mathbf{x}_{t+1} - \mathbf{x}_t\rangle + \frac{L}{2}\|\mathbf{x}_{t+1} - \mathbf{x}_t\|^2 \\
\le &f(\mathbf{x}_t) - \eta\left\langle \nabla f(\mathbf{x}_t), \mathrm{Sign}\left(\frac{1}{n}\sum_{j=1}^{n}\mathrm{S}_R(\mathbf{v}_t^j)\right)\right\rangle + \frac{\eta^2 Ld}{2} \\
&- \eta\left\langle \nabla f(\mathbf{x}_t), \mathrm{Sign}(\nabla f(\mathbf{x}_t))\right\rangle + \frac{\eta^2 Ld}{2} \\
= &f(\mathbf{x}_t) + \eta\left\langle \nabla f(\mathbf{x}_t), \mathrm{Sign}(\nabla f(\mathbf{x}_t)) - \mathrm{Sign}\left(\frac{1}{n}\sum_{j=1}^{n}\mathrm{S}_R(\mathbf{v}_t^j)\right)\right\rangle - \eta\left\|\nabla f(\mathbf{x}_t)\right\|_1 + \frac{\eta^2 Ld}{2} \\
\le &f(\mathbf{x}_t) + 2\eta R\sqrt{d}\left\|\frac{\nabla f(\mathbf{x}_t)}{R} - \frac{1}{n}\sum_{j=1}^{n}\mathrm{S}_R(\mathbf{v}_t^j)\right\| - \eta\left\|\nabla f(\mathbf{x}_t)\right\|_1 + \frac{\eta^2 Ld}{2},
\end{aligned}
$$

$$(10)$$

where the last inequality is because of

$$
\begin{aligned}
&\left\langle \nabla f(\mathbf{x}_t), \mathrm{Sign}(\nabla f(\mathbf{x}_t)) - \mathrm{Sign}\left(\frac{1}{n}\sum_{j=1}^{n}\mathrm{S}_R(\mathbf{v}_t^j)\right)\right\rangle \\
= &\sum_{i=1}^{d}\left\langle [\nabla f(\mathbf{x}_t)]_i, \mathrm{Sign}([\nabla f(\mathbf{x}_t)]_i) - \mathrm{Sign}\left(\left[\frac{1}{n}\sum_{j=1}^{n}\mathrm{S}_R(\mathbf{v}_t^j)\right]_i\right)\right\rangle \\
\le &\sum_{i=1}^{d}2R\,|[\nabla f(\mathbf{x}_t)]_i/R|\cdot\mathbb{I}\left(\mathrm{Sign}([\nabla f(\mathbf{x}_t)]_i) \ne \mathrm{Sign}\left(\left[\frac{1}{n}\sum_{j=1}^{n}\mathrm{S}_R(\mathbf{v}_t^j)\right]_i\right)\right) \\
\le &\sum_{i=1}^{d}2R\left|\frac{[\nabla f(\mathbf{x}_t)]_i}{R} - \left[\frac{1}{n}\sum_{j=1}^{n}\mathrm{S}_R(\mathbf{v}_t^j)\right]_i\right|\cdot\mathbb{I}\left(\mathrm{Sign}([\nabla f(\mathbf{x}_t)]_i) \ne \mathrm{Sign}\left(\left[\frac{1}{n}\sum_{j=1}^{n}\mathrm{S}_R(\mathbf{v}_t^j)\right]_i\right)\right) \\
\le &\sum_{i=1}^{d}2R\left|\frac{[\nabla f(\mathbf{x}_t)]_i}{R} - \left[\frac{1}{n}\sum_{j=1}^{n}\mathrm{S}_R(\mathbf{v}_t^j)\right]_i\right| \\
= &2R\left\|\frac{\nabla f(\mathbf{x}_t)}{R} - \frac{1}{n}\sum_{j=1}^{n}\mathrm{S}_R(\mathbf{v}_t^j)\right\|_1 \le 2R\sqrt{d}\left\|\frac{\nabla f(\mathbf{x}_t)}{R} - \frac{1}{n}\sum_{j=1}^{n}\mathrm{S}_R(\mathbf{v}_t^j)\right\|.
\end{aligned}
$$

$$(11)$$

Rearranging and taking the expectation over equation (10), we have:

$$\mathbb{E}\left[f(\mathbf{x}_{t+1}) - f(\mathbf{x}_t)\right]$$

$$\leq 2\eta R\sqrt{d}\mathbb{E}\left[\left\|\frac{\nabla f(\mathbf{x}_t)}{R} - \frac{1}{n}\sum_{j=1}^{n}\mathrm{S}_R(\mathbf{v}_t^j)\right\|\right] - \eta\mathbb{E}\left[\|\nabla f(\mathbf{x}_t)\|_1\right] + \frac{\eta^2 Ld}{2}$$

$$\leq 2\eta R\sqrt{d}\mathbb{E}\left[\left\|\frac{\nabla f(\mathbf{x}_t)}{R} - \frac{1}{nR}\sum_{j=1}^{n}\mathbf{v}_t^j\right\|\right] + 2\eta R\sqrt{d}\mathbb{E}\left[\left\|\frac{1}{n}\sum_{j=1}^{n}\left(\mathrm{S}_R(\mathbf{v}_t^j) - \frac{\mathbf{v}_t^j}{R}\right)\right\|\right]$$

$$- \eta\mathbb{E}\left[\|\nabla f(\mathbf{x}_t)\|_1\right] + \frac{\eta^2 Ld}{2}$$

$$\leq 2\eta\sqrt{d}\mathbb{E}\left[\left\|\nabla f(\mathbf{x}_t) - \frac{1}{n}\sum_{j=1}^{n}\mathbf{v}_t^j\right\|\right] + 2\eta R\sqrt{d}\sqrt{\mathbb{E}\left[\left\|\frac{1}{n}\sum_{j=1}^{n}\left(\mathrm{S}_R(\mathbf{v}_t^j) - \frac{\mathbf{v}_t^j}{R}\right)\right\|^2\right]}$$

$$- \eta\mathbb{E}\left[\|\nabla f(\mathbf{x}_t)\|_1\right] + \frac{\eta^2 Ld}{2} \tag{12}$$

$$\leq 2\eta\sqrt{d}\mathbb{E}\left[\left\|\nabla f(\mathbf{x}_t) - \frac{1}{n}\sum_{j=1}^{n}\mathbf{v}_t^j\right\|\right] + 2\eta R\sqrt{d}\sqrt{\frac{1}{n^2}\sum_{j=1}^{n}\mathbb{E}\left[\left\|\left(\mathrm{S}_R(\mathbf{v}_t^j) - \frac{\mathbf{v}_t^j}{R}\right)\right\|^2\right]}$$

$$- \eta\mathbb{E}\left[\|\nabla f(\mathbf{x}_t)\|_1\right] + \frac{\eta^2 Ld}{2}$$

$$\leq 2\eta\sqrt{d}\mathbb{E}\left[\left\|\nabla f(\mathbf{x}_t) - \frac{1}{n}\sum_{j=1}^{n}\mathbf{v}_t^j\right\|\right] + 2\eta R\sqrt{d}\sqrt{\frac{1}{n^2}\sum_{j=1}^{n}\mathbb{E}\left[\left\|\mathrm{S}_R(\mathbf{v}_t^j)\right\|^2\right]}$$

$$- \eta\mathbb{E}\left[\|\nabla f(\mathbf{x}_t)\|_1\right] + \frac{\eta^2 Ld}{2}$$

$$\leq 2\eta\sqrt{d}\mathbb{E}\left[\left\|\nabla f(\mathbf{x}_t) - \frac{1}{n}\sum_{j=1}^{n}\mathbf{v}_t^j\right\|\right] + \frac{2\eta dR}{\sqrt{n}} - \eta\mathbb{E}\left[\|\nabla f(\mathbf{x}_t)\|_1\right] + \frac{\eta^2 Ld}{2},$$

where the third inequality is due to the fact that $(\mathbb{E}\left[X\right])^2 \leq \mathbb{E}\left[X^2\right]$, and the forth inequality is because of $\mathbb{E}\left[S\left(\mathbf{v}_t^j\right)\right] = \frac{\mathbf{v}_t^j}{R}$, as well as the $S$ operation in each node is independent.

Rearranging the terms and summing up, we have:

$$\frac{1}{T}\sum_{i=1}^{T}\mathbb{E}\left[\|\nabla f(\mathbf{x}_t)\|_1\right] \leq \frac{\Delta_f}{\eta T} + 2\sqrt{d}\mathbb{E}\left[\frac{1}{T}\sum_{i=1}^{T}\left\|\nabla f(\mathbf{x}_t) - \frac{1}{n}\sum_{j=1}^{n}\mathbf{v}_t^j\right\|\right] + \frac{2dR}{\sqrt{n}} + \frac{\eta Ld}{2}$$

$$\leq \frac{\Delta_f}{\eta T} + 2\sqrt{d}\sqrt{\mathbb{E}\left[\frac{1}{T}\sum_{i=1}^{T}\left\|\nabla f(\mathbf{x}_t) - \frac{1}{n}\sum_{j=1}^{n}\mathbf{v}_t^j\right\|^2\right]} + \frac{2dR}{\sqrt{n}} + \frac{\eta Ld}{2},$$

where the last inequality is due to Jensen's inequality.

For each worker $j$, we have the following according to the definition of $\mathbf{v}_t^j$:

$$\mathbf{v}_{t+1}^j - \nabla f_j(\mathbf{x}_{t+1}) = (1-\beta)\left(\mathbf{v}_t^j - \nabla f_j(\mathbf{x}_t)\right) + \beta\left(\nabla f_j(\mathbf{x}_{t+1}; \xi_{t+1}^j) - \nabla f_j(\mathbf{x}_{t+1})\right)$$

$$+ (1-\beta)\left(\nabla f_j(\mathbf{x}_{t+1}; \xi_{t+1}^j) - \nabla f_j(\mathbf{x}_t; \xi_{t+1}^j) + \nabla f_j(\mathbf{x}_t) - \nabla f_j(\mathbf{x}_{t+1})\right).$$

Summing over $\{n\}$ and noting that $\nabla f(\mathbf{x}) = \frac{1}{n}\sum_{j=1}^{n}\nabla f_j(\mathbf{x})$, we can obtain:

$$\frac{1}{n}\sum_{j=1}^{n}\mathbf{v}_{t+1}^{j} - \nabla f(\mathbf{x}_{t+1}) = \frac{1}{n}\sum_{j=1}^{n}\left(\mathbf{v}_{t+1}^{j} - \nabla f_j(\mathbf{x}_{t+1})\right)$$

$$=(1-\beta)\frac{1}{n}\sum_{j=1}^{n}\left(\mathbf{v}_{t}^{j} - \nabla f_j(\mathbf{x}_t)\right) + \beta\frac{1}{n}\sum_{j=1}^{n}\left(\nabla f_j(\mathbf{x}_{t+1};\xi_{t+1}^{j}) - \nabla f_j(\mathbf{x}_{t+1})\right)$$

$$+(1-\beta)\frac{1}{n}\sum_{j=1}^{n}\left(\nabla f_j(\mathbf{x}_{t+1};\xi_{t+1}^{j}) - \nabla f_j(\mathbf{x}_t;\xi_{t+1}^{j}) + \nabla f_j(\mathbf{x}_t) - \nabla f_j(\mathbf{x}_{t+1})\right).$$

Then we have

$$\mathbb{E}\left[\left\|\frac{1}{n}\sum_{j=1}^{n}\mathbf{v}_{t+1}^{j} - \nabla f(\mathbf{x}_{t+1})\right\|^{2}\right]$$

$$\leq(1-\beta)^{2}\mathbb{E}\left[\left\|\frac{1}{n}\sum_{j=1}^{n}\left(\mathbf{v}_{t}^{j} - \nabla f_j(\mathbf{x}_t)\right)\right\|^{2}\right] + 2\beta^{2}\frac{1}{n^{2}}\sum_{j=1}^{n}\mathbb{E}\left[\left\|\nabla f_j(\mathbf{x}_{t+1};\xi_{t+1}^{j}) - \nabla f_j(\mathbf{x}_{t+1})\right\|^{2}\right]$$

$$+2(1-\beta)^{2}\frac{1}{n^{2}}\sum_{j=1}^{n}\mathbb{E}\left[\left\|\nabla f_j(\mathbf{x}_{t+1};\xi_{t+1}^{j}) - \nabla f_j(\mathbf{x}_t;\xi_{t+1}^{j})\right\|^{2}\right]$$

$$\leq(1-\beta)\mathbb{E}\left[\left\|\frac{1}{n}\sum_{j=1}^{n}\left(\mathbf{v}_{t}^{j} - \nabla f_j(\mathbf{x}_t)\right)\right\|^{2}\right] + \frac{2\beta^{2}\sigma^{2}}{n} + \frac{2L^{2}}{n}\left\|\mathbf{x}_{t+1} - \mathbf{x}_t\right\|^{2}$$

$$\leq(1-\beta)\mathbb{E}\left[\left\|\frac{1}{n}\sum_{j=1}^{n}\mathbf{v}_{t}^{j} - \nabla f(\mathbf{x}_t)\right\|^{2}\right] + \frac{2\beta^{2}\sigma^{2}}{n} + \frac{2L^{2}\eta^{2}d}{n}.$$

By summing up and rearranging, we observe

$$\mathbb{E}\left[\frac{1}{T}\sum_{t=1}^{T}\left\|\frac{1}{n}\sum_{j=1}^{n}\mathbf{v}_{t}^{j} - \nabla f(\mathbf{x}_t)\right\|^{2}\right] \leq \frac{\mathbb{E}\left[\left\|\frac{1}{n}\sum_{j=1}^{n}\mathbf{v}_{1}^{j} - \nabla f(\mathbf{x}_1)\right\|^{2}\right]}{\beta T} + \frac{2\sigma^{2}\beta}{n} + \frac{2L^{2}\eta^{2}d}{n\beta} \quad (13)$$

$$\leq \frac{\sigma^{2}}{n\beta T} + \frac{2\sigma^{2}\beta}{n} + \frac{2L^{2}\eta^{2}d}{n\beta}.$$

Finally, by setting $\beta = \frac{1}{2}$ and $\eta = \mathcal{O}\left(T^{-1/2}d^{-1/2}\right)$, we ensure that

$$\frac{1}{T}\sum_{i=1}^{T}\|\nabla f(\mathbf{x}_t)\|_1 \leq \frac{\Delta_f}{\eta T} + \frac{2dR}{\sqrt{n}} + \frac{\eta Ld}{2} + 2\sqrt{d}\sqrt{\mathbb{E}\left[\frac{1}{T}\sum_{i=1}^{T}\left\|\nabla f(\mathbf{x}_t) - \frac{1}{n}\sum_{j=1}^{n}\mathbf{v}_{t}^{j}\right\|^{2}\right]}$$

$$\leq \frac{\Delta_f}{\eta T} + \frac{2dR}{\sqrt{n}} + \frac{\eta Ld}{2} + 2\sqrt{d}\sqrt{\frac{\sigma^{2}}{n\beta T} + \frac{2\sigma^{2}\beta}{n} + \frac{2L^{2}\eta^{2}d}{n\beta}}$$

$$= \mathcal{O}\left(\frac{d^{1/2}}{T^{1/2}} + \frac{d}{n^{1/2}}\right).$$

# D   Proof of Theorem 4

Due to the fact that the overall objective function $f(\mathbf{x})$ is $L$-smooth, we have the following:

$$f(\mathbf{x}_{t+1}) \leq f(\mathbf{x}_t) + \langle \nabla f(\mathbf{x}_t), \mathbf{x}_{t+1} - \mathbf{x}_t \rangle + \frac{L}{2}\|\mathbf{x}_{t+1} - \mathbf{x}_t\|^2$$

$$\leq f(\mathbf{x}_t) - \eta \left\langle \nabla f(\mathbf{x}_t), \mathrm{S}_1\left(\frac{1}{n}\sum_{j=1}^n \mathrm{S}_G(\hat{\mathbf{v}}_t^j)\right) \right\rangle + \frac{\eta^2 Ld}{2}$$

$$= f(\mathbf{x}_t) + \eta \left\langle \nabla f(\mathbf{x}_t), \nabla f(\mathbf{x}_t) - \mathrm{S}_1\left(\frac{1}{n}\sum_{j=1}^n \mathrm{S}_G(\hat{\mathbf{v}}_t^j)\right) \right\rangle - \eta\|\nabla f(\mathbf{x}_t)\|^2 + \frac{\eta^2 Ld}{2}.$$

Taking expectations leads to:

$$\mathbb{E}\left[f(\mathbf{x}_{t+1}) - f(\mathbf{x}_t)\right]$$

$$\leq \eta \mathbb{E}\left[\left\langle \nabla f(\mathbf{x}_t), \frac{1}{G}\nabla f(\mathbf{x}_t) - \mathrm{S}_1\left(\frac{1}{n}\sum_{j=1}^n \mathrm{S}_G(\hat{\mathbf{v}}_t^j)\right) \right\rangle\right] - \frac{\eta}{G}\mathbb{E}\left[\|\nabla f(\mathbf{x}_t)\|^2\right] + \frac{\eta^2 Ld}{2}$$

$$\leq \eta \mathbb{E}\left[\left\langle \nabla f(\mathbf{x}_t), \frac{1}{G}\nabla f(\mathbf{x}_t) - \frac{1}{n}\sum_{j=1}^n \mathrm{S}_G(\hat{\mathbf{v}}_t^j) \right\rangle\right] - \frac{\eta}{G}\mathbb{E}\left[\|\nabla f(\mathbf{x}_t)\|^2\right] + \frac{\eta^2 Ld}{2}$$

$$\leq \eta \mathbb{E}\left[\left\langle \nabla f(\mathbf{x}_t), \frac{1}{G}\nabla f(\mathbf{x}_t) - \frac{1}{nG}\sum_{j=1}^n \hat{\mathbf{v}}_t^j \right\rangle\right] - \frac{\eta}{G}\mathbb{E}\left[\|\nabla f(\mathbf{x}_t)\|^2\right] + \frac{\eta^2 Ld}{2} \qquad (14)$$

$$\leq \eta \mathbb{E}\left[\frac{1}{2G}\|\nabla f(\mathbf{x}_t)\|^2 + \frac{1}{2G}\left\|\nabla f(\mathbf{x}_t) - \frac{1}{n}\sum_{j=1}^n \hat{\mathbf{v}}_t^j\right\|^2\right] - \frac{\eta}{G}\mathbb{E}\left[\|\nabla f(\mathbf{x}_t)\|^2\right] + \frac{\eta^2 Ld}{2}$$

$$\leq \frac{\eta}{2G}\mathbb{E}\left[\left\|\nabla f(\mathbf{x}_t) - \frac{1}{n}\sum_{j=1}^n \hat{\mathbf{v}}_t^j\right\|^2\right] - \frac{\eta}{2G}\mathbb{E}\left[\|\nabla f(\mathbf{x}_t)\|^2\right] + \frac{\eta^2 Ld}{2}$$

Rearranging the terms and summing up:

$$\frac{1}{T}\sum_{i=1}^T \mathbb{E}\left[\|\nabla f(\mathbf{x}_t)\|^2\right] \leq \frac{2\Delta_f G}{\eta T} + \mathbb{E}\left[\frac{1}{T}\sum_{i=1}^T\left\|\nabla f(\mathbf{x}_t) - \frac{1}{n}\sum_{j=1}^n \hat{\mathbf{v}}_t^j\right\|^2\right] + \eta LdG$$

$$\leq \frac{2\Delta_f G}{\eta T} + \mathbb{E}\left[\frac{1}{n}\sum_{j=1}^n \frac{1}{T}\sum_{i=1}^T\left\|\nabla f_j(\mathbf{x}_t) - \hat{\mathbf{v}}_t^j\right\|^2\right] + \eta LdG$$

$$\leq \frac{2\Delta_f G}{\eta T} + \mathbb{E}\left[\frac{1}{n}\sum_{j=1}^n \frac{1}{T}\sum_{i=1}^T\left\|\nabla f_j(\mathbf{x}_t) - \Pi_G\left[\mathbf{v}_t^j\right]\right\|^2\right] + \eta LdG$$

$$\leq \frac{2\Delta_f G}{\eta T} + \mathbb{E}\left[\frac{1}{n}\sum_{j=1}^n \frac{1}{T}\sum_{i=1}^T\left\|\nabla f_j(\mathbf{x}_t) - \mathbf{v}_t^j\right\|^2\right] + \eta LdG.$$

where the last inequality is due to the non-expansive property of the projection operation.

For each worker $j$, according to the definition of $\mathbf{v}_t^j$, we have:

$$\mathbf{v}_{t+1}^j - \nabla f_j(\mathbf{x}_{t+1}) = (1-\beta)\left(\mathbf{v}_t^j - \nabla f_j(\mathbf{x}_t)\right) + \beta\left(\nabla f_j(\mathbf{x}_{t+1}; \xi_{t+1}^j) - \nabla f_j(\mathbf{x}_{t+1})\right)$$

$$+ (1-\beta)\left(\nabla f_j(\mathbf{x}_{t+1}; \xi_{t+1}^j) - \nabla f_j(\mathbf{x}_t; \xi_{t+1}^j) + \nabla f_j(\mathbf{x}_t) - \nabla f_j(\mathbf{x}_{t+1})\right).$$

Then we have

$$\mathbb{E}\left[\left\|\mathbf{v}_{t+1}^j - \nabla f_j(\mathbf{x}_{t+1})\right\|^2\right]$$

$$\leq(1-\beta)^2\mathbb{E}\left[\left\|\mathbf{v}_t^j - \nabla f_j(\mathbf{x}_t)\right\|^2\right] + 2\beta^2\mathbb{E}\left[\left\|\left(\nabla f_j(\mathbf{x}_{t+1};\xi_{t+1}^j) - \nabla f_j(\mathbf{x}_{t+1})\right)\right\|^2\right]$$

$$+ 2(1-\beta)^2\mathbb{E}\left[\left\|\left(\nabla f_j(\mathbf{x}_{t+1};\xi_{t+1}^j) - \nabla f_j(\mathbf{x}_t;\xi_{t+1}^j)\right)\right\|^2\right]$$

$$\leq(1-\beta)\mathbb{E}\left[\left\|\mathbf{v}_t^j - \nabla f_j(\mathbf{x}_t)\right\|^2\right] + 2\beta^2\sigma^2 + 2L^2\|\mathbf{x}_{t+1} - \mathbf{x}_t\|^2$$

$$\leq(1-\beta)\mathbb{E}\left[\left\|\mathbf{v}_t^j - \nabla f_j(\mathbf{x}_t)\right\|^2\right] + 2\beta^2\sigma^2 + 2L^2\eta^2 d.$$

As a result, we know that

$$\mathbb{E}\left[\frac{1}{n}\sum_{j=1}^n\frac{1}{T}\sum_{t=1}^T\left\|\mathbf{v}_t^j - \nabla f_j(\mathbf{x}_t)\right\|^2\right] \leq \frac{\sigma^2}{\beta T} + 2\sigma^2\beta + \frac{2L^2\eta^2 d}{\beta}.$$

Finally, combining the above and setting that $\beta = \mathcal{O}\left(\frac{1}{T^{1/2}}\right)$, $\eta = \mathcal{O}\left(\frac{1}{d^{1/2}T^{1/2}}\right)$, we obtain the final bound:

$$\mathbb{E}\left[\frac{1}{T}\sum_{i=1}^T\|\nabla f(\mathbf{x}_t)\|\right] \leq \sqrt{\mathbb{E}\left[\frac{1}{T}\sum_{i=1}^T\|\nabla f(\mathbf{x}_t)\|^2\right]}$$

$$\leq \sqrt{\frac{2\Delta_f G}{\eta T} + \eta L d G + \frac{\sigma^2}{\beta T} + 2\sigma^2\beta + \frac{2L^2\eta^2 d}{\beta}}$$

$$\leq \mathcal{O}\left(\sqrt{\frac{d^{1/2}\left(\Delta_f + L\right) + \sigma^2 + L^2}{T^{1/2}}}\right)$$

$$= \mathcal{O}\left(\frac{d^{1/4}}{T^{1/4}}\right).$$

# E   Results under weaker assumptions

In this section, we demonstrate that our proposed methods can maintain similar convergence rates under less stringent assumptions — expected $\alpha$-symmetric generalized-smoothness [Chen et al., 2023] and affine variance [Faw et al., 2022]. We first detail these relaxed assumptions below.

**Assumption 1′** *(Expected $\alpha$-symmetric generalized-smoothness)*

$$\mathbb{E}_\xi\left[\|\nabla f(\mathbf{x};\xi) - \nabla f(\mathbf{y};\xi)\|^2\right] \leq \|\mathbf{x}-\mathbf{y}\|^2\mathbb{E}_\xi\left[\left(L_0 + L_1\max_{\theta\in[0,1]}\|\nabla f(\mathbf{x}_\theta;\xi)\|^\alpha\right)^2\right],$$

*where $\mathbf{x}_\theta := \theta\mathbf{x} + (1-\theta)\mathbf{y}$, and $0 \leq \alpha \leq 1$.*

**Assumption 2′** *(Affine variance)*

$$\mathbb{E}_\xi\left[\|\nabla f(\mathbf{x};\xi) - \nabla f(\mathbf{x})\|^2\right] \leq \Gamma^2\|\nabla f(\mathbf{x})\|^2 + \Lambda^2.$$

**Remark:** Assumption 1′ can be reduced to standard average smoothness (Assumption 1) when $L_1 = 0$. Note that $\alpha$-symmetric generalized-smooth functions not only include asymmetric and Hessian-based generalized-smooth functions, but also contain high-order polynomials and exponential functions [Chen et al., 2023]. Moreover, affine variance is also weaker than Assumption 2 and can be reduced to it when $\Gamma = 0$.

We then demonstrate that these relaxed conditions are sufficient for our algorithms to achieve the same convergence rate.

**Theorem 5** *Under Assumptions 1' and 2', by setting that $\beta = \mathcal{O}(\frac{d^{1/3}}{T^{2/3}})$, $\eta = \mathcal{O}(\frac{1}{d^{1/6}T^{2/3}})$, $B_0 = \mathcal{O}(1)$, $B_1 = \mathcal{O}(d)$, and denoting $N$ as the samples used, our SSVR method guarantees:*

$$\mathbb{E}\left[\|\nabla f(\mathbf{x}_\tau)\|_1\right] \leq \mathcal{O}\left(\frac{d^{1/2}}{N^{1/3}}\right).$$

Furthermore, we introduce the following relaxed assumption for the finite-sum problem.

**Assumption 3'** *(Generalized smoothness) For each $i \in \{1, 2, \cdots, m\}$, we have*

$$\|\nabla f_i(\mathbf{x}) - \nabla f_i(\mathbf{y})\| \leq \|\mathbf{x} - \mathbf{y}\| \left(L_0 + L_1 \max_{\theta \in [0,1]} \|\nabla f(\mathbf{x}_\theta)\|^\alpha\right),$$

*where $\mathbf{x}_\theta := \theta \mathbf{x} + (1 - \theta)\mathbf{y}$, and $0 \leq \alpha \leq 1$.*

This assumption is weaker than the standard Assumption 3. We validate that our SSVR-FS algorithm can still achieve similar convergence under this relaxed condition.

**Theorem 6** *Under Assumption 3', by setting $\eta = \mathcal{O}(\min\{\frac{1}{m^{1/4}d^{1/2}T^{1/2}}, \frac{1}{md}\})$, $\beta = \mathcal{O}(\frac{1}{m})$, and $I = m$, our SSVR-FS algorithm ensures:*

$$\mathbb{E}\left[\|\nabla F(\mathbf{x}_\varphi)\|_1\right] \leq \mathcal{O}\left(\frac{m^{1/4}d^{1/2}}{T^{1/2}} + \frac{md}{T}\right).$$

**Remark:** When the iteration number $T$ is large, the dominant term becomes $\mathcal{O}(m^{1/4}d^{1/2}T^{-1/2})$, which aligns with the results in Theorem 2.

### E.1 Proof of Theorem 5

We first present some useful tools for analysis. According to Proposition 4 in Chen et al. [2023], Assumption 1' leads to the following lemmas.

**Lemma 1** *For $\alpha \in (0, 1)$, we have:*

$$\begin{aligned}
f(\mathbf{x}_{t+1}) \leq & f(\mathbf{x}_t) + \langle \nabla f(\mathbf{x}_t), \mathbf{x}_{t+1} - \mathbf{x}_t \rangle \\
& + \frac{1}{2}\|\mathbf{x}_{t+1} - \mathbf{x}_t\|^2 \left(K_0 + K_1\|\nabla f(\mathbf{x}_t)\|^\alpha + 2K_2\|\mathbf{x}_{t+1} - \mathbf{x}_t\|^{\frac{\alpha}{1-\alpha}}\right),
\end{aligned}$$

*where $K_0 := L_0\left(2^{\frac{\alpha^2}{1-\alpha}} + 1\right)$, $K_1 := L_1 \cdot 2^{\frac{\alpha^2}{1-\alpha}} \cdot 3^\alpha$, $K_2 := L_1^{\frac{1}{1-\alpha}} \cdot 2^{\frac{\alpha^2}{1-\alpha}} \cdot 3^\alpha(1-\alpha)^{\frac{\alpha}{1-\alpha}}$.*
*For $\alpha = 1$, we also have:*

$$\begin{aligned}
f(\mathbf{x}_{t+1}) \leq & f(\mathbf{x}_t) + \langle \nabla f(\mathbf{x}_t), \mathbf{x}_{t+1} - \mathbf{x}_t \rangle \\
& + \frac{1}{2}\|\mathbf{x}_{t+1} - \mathbf{x}_t\|^2 \left(L_0 + L_1\|\nabla f(\mathbf{x}_t)\|\right) \exp\left(L_1\|\mathbf{x}_{t+1} - \mathbf{x}_t\|\right),
\end{aligned}$$

Similarly, according to the Proposition 4 in Chen et al. [2023], we have the following guarantees.

**Lemma 2** *For $\alpha \in (0, 1)$, we have:*

$$\mathbb{E}_\xi\|\nabla f(\mathbf{x}; \xi) - \nabla f(\mathbf{y}; \xi)\|^2 \leq \|\mathbf{x} - \mathbf{y}\|^2\left(\overline{K}_0 + \overline{K}_1\mathbb{E}_\xi\|\nabla f(\mathbf{y}; \xi)\|^\alpha + \overline{K}_2\|\mathbf{x} - \mathbf{y}\|^{\frac{\alpha}{1-\alpha}}\right)^2.$$

*where $\overline{K}_0 = 2^{\frac{2-\alpha}{1-\alpha}}L_0$, $\overline{K}_1 = 2^{\frac{2-\alpha}{1-\alpha}}L_1$, $\overline{K}_2 = (5L_1)^{\frac{1}{1-\alpha}}$.*
*For $\alpha = 1$, we also have:*

$$\mathbb{E}_\xi\|\nabla f(\mathbf{x}; \xi) - \nabla f(\mathbf{y}; \xi)\|^2 \leq 2\|\mathbf{x} - \mathbf{y}\|^2(L_0^2 + 2L_1^2\mathbb{E}_\xi\|\nabla f(\mathbf{y}; \xi)\|^2)\exp(12L_1^2\|\mathbf{x} - \mathbf{y}\|^2).$$

Then, we can begin our proof. For $\alpha \in (0, 1)$, according to Lemma 1, by setting $\eta \leq d^{-\frac{1}{2}}$, we have:

$$f(\mathbf{x}_{t+1})$$
$$\leq f(\mathbf{x}_t) + \langle \nabla f(\mathbf{x}_t), \mathbf{x}_{t+1} - \mathbf{x}_t \rangle$$
$$+ \frac{1}{2} \|\mathbf{x}_{t+1} - \mathbf{x}_t\|^2 \left( K_0 + K_1 \|\nabla f(\mathbf{x}_t)\|^\alpha + 2K_2 \|\mathbf{x}_{t+1} - \mathbf{x}_t\|^{\frac{\alpha}{1-\alpha}} \right)$$
$$\leq f(\mathbf{x}_t) + \langle \nabla f(\mathbf{x}_t), -\eta \operatorname{Sign}(\mathbf{v}_t) \rangle$$
$$+ \frac{1}{2} \|\mathbf{x}_{t+1} - \mathbf{x}_t\|^2 \left( K_0 + K_1 \left( 1 + \|\nabla f(\mathbf{x}_t)\| \right) + 2K_2 \right)$$
$$\leq f(\mathbf{x}_t) + \eta \langle \nabla f(\mathbf{x}_t), \operatorname{Sign}(\nabla f(\mathbf{x}_t)) - \operatorname{Sign}(\mathbf{v}_t) \rangle - \eta \langle \nabla f(\mathbf{x}_t), \operatorname{Sign}(\nabla f(\mathbf{x}_t)) \rangle$$
$$+ \frac{1}{2} \|\mathbf{x}_{t+1} - \mathbf{x}_t\|^2 \left( K_0 + K_1 \left( 1 + \|\nabla f(\mathbf{x}_t)\| \right) + 2K_2 \right)$$
$$= f(\mathbf{x}_t) + \eta \langle \nabla f(\mathbf{x}_t), \operatorname{Sign}(\nabla f(\mathbf{x}_t)) - \operatorname{Sign}(\mathbf{v}_t) \rangle - \eta \|\nabla f(\mathbf{x}_t)\|_1$$
$$+ \frac{1}{2} \|\mathbf{x}_{t+1} - \mathbf{x}_t\|^2 \left( (K_0 + K_1 + 2K_2) + K_1 \|\nabla f(\mathbf{x}_t)\| \right)$$
$$\leq f(\mathbf{x}_t) + 2\eta \sqrt{d} \|\nabla f(\mathbf{x}_t) - \mathbf{v}_t\| - \eta \|\nabla f(\mathbf{x}_t)\|_1 + \frac{\eta^2 d}{2} \left( (K_0 + K_1 + 2K_2) + K_1 \|\nabla f(\mathbf{x}_t)\| \right),$$

where the second inequality is due to the fact that $\alpha < 1$ and $\|\mathbf{x}_{t+1} - \mathbf{x}_t\|^2 \leq \eta^2 d \leq 1$.

Rearranging and summing up, we then have

$$\mathbb{E} \left[ \frac{1}{T} \sum_{t=1}^T \|\nabla f(\mathbf{x}_t)\|_1 \right] \leq \frac{\Delta_f}{\eta T} + 2\sqrt{d} \cdot \mathbb{E} \left[ \frac{1}{T} \sum_{t=1}^T \|\nabla f(\mathbf{x}_t) - \mathbf{v}_t\| \right]$$
$$+ \frac{\eta d (K_0 + K_1 + 2K_2)}{2} + \frac{\eta d K_1}{2T} \mathbb{E} \left[ \sum_{t=1}^T \|\nabla f(\mathbf{x}_t)\| \right].$$

By setting $\eta \leq \min\{\frac{1}{\sqrt{d}}, \frac{1}{dK_1}\}$, we can get

$$\mathbb{E} \left[ \frac{1}{T} \sum_{t=1}^T \|\nabla f(\mathbf{x}_t)\|_1 \right] \leq \frac{2\Delta_f}{\eta T} + 4\sqrt{d} \cdot \mathbb{E} \left[ \frac{1}{T} \sum_{t=1}^T \|\nabla f(\mathbf{x}_t) - \mathbf{v}_t\| \right] + \eta d (K_0 + K_1 + 2K_2). \tag{15}$$

For $\alpha = 1$, according to Lemma 1 and setting $\eta \leq \frac{1}{L_1 \sqrt{d}}$, we have

$$f(\mathbf{x}_{t+1})$$
$$\leq f(\mathbf{x}_t) + \langle \nabla f(\mathbf{x}_t), \mathbf{x}_{t+1} - \mathbf{x}_t \rangle + \frac{1}{2} \|\mathbf{x}_{t+1} - \mathbf{x}_t\|^2 \left( L_0 + L_1 \|\nabla f(\mathbf{x}_t)\| \right) \exp \left( L_1 \|\mathbf{x}_{t+1} - \mathbf{x}_t\| \right)$$
$$\leq f(\mathbf{x}_t) + \langle \nabla f(\mathbf{x}_t), \mathbf{x}_{t+1} - \mathbf{x}_t \rangle + \frac{\exp(1)}{2} \eta^2 d \left( L_0 + L_1 \|\nabla f(\mathbf{x}_t)\| \right)$$
$$\leq f(\mathbf{x}_t) + 2\eta \sqrt{d} \|\nabla f(\mathbf{x}_t) - \mathbf{v}_t\| - \eta \|\nabla f(\mathbf{x}_t)\|_1 + \frac{3\eta^2 d}{2} \left( L_0 + L_1 \|\nabla f(\mathbf{x}_t)\| \right),$$

where the second inequality is due to $L_1 \|\mathbf{x}_{t+1} - \mathbf{x}_t\| \leq 1$, and others follow the previous proof.

Rearranging and summing up, we then have

$$\mathbb{E} \left[ \frac{1}{T} \sum_{t=1}^T \|\nabla f(\mathbf{x}_t)\|_1 \right]$$
$$\leq \frac{\Delta_f}{\eta T} + 2\sqrt{d} \cdot \mathbb{E} \left[ \frac{1}{T} \sum_{t=1}^T \|\nabla f(\mathbf{x}_t) - \mathbf{v}_t\| \right] + \frac{3\eta d L_0}{2} + \frac{3\eta d L_1}{2T} \mathbb{E} \left[ \sum_{t=1}^T \|\nabla f(\mathbf{x}_t)\| \right].$$

By setting $\eta \leq \frac{1}{3dL_1}$, we can get

$$\mathbb{E} \left[ \frac{1}{T} \sum_{t=1}^T \|\nabla f(\mathbf{x}_t)\|_1 \right] \leq \frac{2\Delta_f}{\eta T} + 4\sqrt{d} \cdot \mathbb{E} \left[ \frac{1}{T} \sum_{t=1}^T \|\nabla f(\mathbf{x}_t) - \mathbf{v}_t\| \right] + 3\eta d L_0. \tag{16}$$

Then we begin to bound the term $\mathbb{E}\left[\frac{1}{T}\sum_{t=1}^{T}\|\nabla f(\mathbf{x}_t) - \mathbf{v}_t\|\right]$. According to the definition of $\mathbf{v}_t$, we have:

$$
\begin{aligned}
\mathbf{v}_t - \nabla f(\mathbf{x}_t) =& (1-\beta)\left(\mathbf{v}_{t-1} - \nabla f(\mathbf{x}_{t-1})\right) + \beta\left(\frac{1}{B_1}\sum_{k=1}^{B_1}\nabla f(\mathbf{x}_t;\xi_t^k) - \nabla f(\mathbf{x}_t)\right) \\
& + (1-\beta)\left(\frac{1}{B_1}\sum_{k=1}^{B_1}\left(\nabla f(\mathbf{x}_t;\xi_t^k) - \nabla f(\mathbf{x}_{t-1};\xi_t^k)\right) + \nabla f(\mathbf{x}_{t-1}) - \nabla f(\mathbf{x}_t)\right).
\end{aligned}
$$

Denote that $G_t = \left(\frac{1}{B_1}\sum_{k=1}^{B_1}\left(\nabla f(\mathbf{x}_t;\xi_t^k) - \nabla f(\mathbf{x}_{t-1};\xi_t^k)\right) + \nabla f(\mathbf{x}_{t-1}) - \nabla f(\mathbf{x}_t)\right)$ and $\Delta_t = \frac{1}{B_1}\sum_{k=1}^{B_1}\left(\nabla f(\mathbf{x}_t;\xi_t^k) - \nabla f(\mathbf{x}_t)\right)$. By summing up, we have

$$
\begin{aligned}
& \mathbf{v}_t - \nabla f(\mathbf{x}_t) \\
=& (1-\beta)(\mathbf{v}_{t-1} - \nabla f(\mathbf{x}_{t-1})) + \beta\Delta_t + (1-\beta)G_t \\
=& \cdots \\
=& (1-\beta)^{t-1}(\mathbf{v}_1 - \nabla f(\mathbf{x}_1)) + \beta\sum_{s=1}^{t}(1-\beta)^{t-s}\Delta_s + (1-\beta)\sum_{s=1}^{t}(1-\beta)^{t-s}G_s.
\end{aligned}
$$

Thus, we can know that

$$
\begin{aligned}
\mathbb{E}\left[\|\mathbf{v}_t - \nabla f(\mathbf{x}_t)\|\right] \leq & (1-\beta)^{t-1}\mathbb{E}\left[\|\mathbf{v}_1 - \nabla f(\mathbf{x}_1)\|\right] \\
& + \beta\mathbb{E}\left[\left\|\sum_{s=1}^{t}(1-\beta)^{t-s}\Delta_s\right\|\right] + (1-\beta)\mathbb{E}\left[\left\|\sum_{s=1}^{t}(1-\beta)^{t-s}G_s\right\|\right].
\end{aligned}
$$

Then we give the following two important lemmas, and their proofs can be found in Appendix E.1.1 and Appendix E.1.2, respectively.

**Lemma 3**

$$
\begin{aligned}
\mathbb{E}\left[\left\|\sum_{s=1}^{t}(1-\beta)^{t-s}\Delta_s\right\|\right] \leq & \mathbb{E}\left[\sqrt{\left\|\sum_{s=1}^{t-r}(1-\beta)^{t-s}\Delta_s\right\|^2 + \frac{1}{B_1}\sum_{s=1}^{r}(1-\beta)^{2s-2}\Lambda^2}\right] \\
& + \sum_{s=t+1-r}^{t}\frac{\Gamma}{\sqrt{B_1}}(1-\beta)^{t-s}\mathbb{E}\left[\|\nabla f(\mathbf{x}_s)\|\right]
\end{aligned}
$$

**Lemma 4**

$$
\begin{aligned}
\mathbb{E}\left[\left\|\sum_{s=1}^{t}(1-\beta)^{t-s}G_s\right\|\right] \leq & \mathbb{E}\left[\sqrt{\left\|\sum_{s=1}^{t-r}(1-\beta)^{t-s}G_s\right\|^2 + \frac{1}{B_1}\sum_{s=1}^{r}2(1-\beta)^{2s-2}\eta^2 L_3^2 d}\right] \\
& + \frac{\sqrt{2d}\eta L_4}{\sqrt{B_1}}\sum_{s=t+1-r}^{t}(1-\beta)^{t-s}\mathbb{E}\left[\|\nabla f(\mathbf{x}_s)\|\right]
\end{aligned}
$$

Using these lemmas and setting $r = t$, we then have

$$\mathbb{E}\left[\left\|\sum_{s=1}^{t}(1-\beta)^{t-s}\Delta_s\right\|\right] \leq \sqrt{\frac{\Lambda^2}{B_1}\sum_{s=1}^{t}(1-\beta)^{2s-2}} + \sum_{s=1}^{t}\frac{\Gamma}{\sqrt{B_1}}(1-\beta)^{t-s}\mathbb{E}\left[\|\nabla f(\mathbf{x}_s)\|\right]$$

$$\leq \frac{\Lambda}{\sqrt{B_1\beta}} + \sum_{s=1}^{t}\frac{\Gamma}{\sqrt{B_1}}(1-\beta)^{t-s}\mathbb{E}\left[\|\nabla f(\mathbf{x}_s)\|\right]$$

$$\mathbb{E}\left[\left\|\sum_{s=1}^{t}(1-\beta)^{t-s}G_s\right\|\right] \leq \sqrt{\frac{2\eta^2 L_3^2 d}{B_1}\sum_{s=1}^{t}(1-\beta)^{2s-2}} + \frac{\sqrt{2d}\eta L_4}{\sqrt{B_1}}\sum_{s=1}^{t}(1-\beta)^{t-s}\mathbb{E}\left[\|\nabla f(\mathbf{x}_s)\|\right]$$

$$\leq \frac{\sqrt{2d}\eta L_3}{\sqrt{B_1\beta}} + \frac{\sqrt{2d}\eta L_4}{\sqrt{B_1}}\sum_{s=1}^{t}(1-\beta)^{t-s}\mathbb{E}\left[\|\nabla f(\mathbf{x}_s)\|\right].$$

Combining above inequalities and setting $\beta = \eta\sqrt{d}$, we derive

$$\mathbb{E}\left[\frac{1}{T}\sum_{t=1}^{T}\|\nabla f(\mathbf{x}_t) - \mathbf{v}_t\|\right]$$

$$\leq \frac{1}{T}\sum_{t=1}^{T}(1-\beta)^{t-1}\mathbb{E}\left[\|\mathbf{v}_1 - \nabla f(\mathbf{x}_1)\|\right] + \beta\frac{1}{T}\sum_{t=1}^{T}\mathbb{E}\left[\left\|\sum_{s=1}^{t}(1-\beta)^{t-s}\Delta_s\right\|\right]$$

$$+ \frac{1}{T}\sum_{t=1}^{T}\mathbb{E}\left[\left\|\sum_{s=1}^{t}(1-\beta)^{t-s}G_s\right\|\right]$$

$$\leq \frac{\sigma}{\sqrt{B_0}}\frac{1}{T}\sum_{t=1}^{T}(1-\beta)^{t-1} + \frac{\Lambda\sqrt{\beta}}{\sqrt{B_1}} + \frac{\sqrt{2d}\eta L_3}{\sqrt{B_1\beta}} + \left(\frac{\beta\Gamma}{\sqrt{B_1}} + \frac{\sqrt{2d}\eta L_4}{\sqrt{B_1}}\right)\frac{1}{T}\sum_{t=1}^{T}\sum_{s=1}^{t}(1-\beta)^{t-s}\mathbb{E}\left[\|\nabla f(\mathbf{x}_s)\|\right]$$

$$\leq \frac{\sigma}{\beta T\sqrt{B_0}} + \frac{\Lambda\sqrt{\beta}}{\sqrt{B_1}} + \frac{\sqrt{2d}\eta L_3}{\sqrt{B_1\beta}} + \left(\frac{\beta\Gamma}{\sqrt{B_1}} + \frac{\sqrt{2d}\eta L_4}{\sqrt{B_1}}\right)\left(\sum_{i=1}^{T}(1-\beta)^i\right)\frac{1}{T}\sum_{t=1}^{T}\mathbb{E}\left[\|\nabla f(\mathbf{x}_t)\|\right]$$

$$\leq \frac{\sigma}{\beta T\sqrt{B_0}} + \frac{\Lambda\sqrt{\beta}}{\sqrt{B_1}} + \frac{\sqrt{2d}\eta L_3}{\sqrt{B_1\beta}} + \left(\frac{\Gamma}{\sqrt{B_1}} + \frac{\sqrt{2d}\eta L_4}{\sqrt{B_1\beta}}\right)\frac{1}{T}\sum_{t=1}^{T}\mathbb{E}\left[\|\nabla f(\mathbf{x}_t)\|\right]$$

$$\leq \frac{\sigma}{\eta T\sqrt{B_0 d}} + \frac{(\Lambda + \sqrt{2}L_3)d^{1/4}\eta^{1/2}}{\sqrt{B_1}} + \left(\frac{\Gamma}{\sqrt{B_1}} + \frac{\sqrt{2}L_4}{\sqrt{B_1}}\right)\frac{1}{T}\sum_{t=1}^{T}\mathbb{E}\left[\|\nabla f(\mathbf{x}_t)\|\right].$$

For $\alpha \in (0,1)$, by setting that

$$B_0 = 1,$$
$$B_1 \geq \max\{256\Gamma^2, 512L_4^2\}d,$$
$$\beta = \frac{d^{1/3}}{T^{2/3}},$$
$$\eta = \frac{1}{d^{1/6}T^{2/3}},$$

and suppose that iteration number

$$T \geq \mathcal{O}(d^2),$$

then we can guarantee

$$\mathbb{E}\left[\frac{1}{T}\sum_{t=1}^{T}\|\nabla f(\mathbf{x}_t)\|_1\right]$$

$$\leq\frac{2\Delta_f}{\eta T}+\eta d(K_0+K_1+2K_2)+\frac{4\sigma}{\eta T\sqrt{B_0}}+\frac{4(\Lambda+\sqrt{2}L_3)d^{3/4}\eta^{1/2}}{\sqrt{B_1}}$$

$$+\left(\frac{4\sqrt{d}\Gamma}{\sqrt{B_1}}+\frac{4\sqrt{2d}L_4}{\sqrt{B_1}}\right)\frac{1}{T}\sum_{t=1}^{T}\mathbb{E}\left[\|\nabla f(\mathbf{x}_t)\|\right]$$

$$\leq\mathcal{O}\left(\frac{d^{1/6}}{T^{1/3}}\right)+\frac{1}{2}\mathbb{E}\left[\frac{1}{T}\sum_{t=1}^{T}\|\nabla f(\mathbf{x}_t)\|\right]$$

$$\leq\mathcal{O}\left(\frac{d^{1/6}}{T^{1/3}}\right)+\frac{1}{2}\mathbb{E}\left[\frac{1}{T}\sum_{t=1}^{T}\|\nabla f(\mathbf{x}_t)\|_1\right],$$

which indicates that

$$\mathbb{E}\left[\frac{1}{T}\sum_{t=1}^{T}\|\nabla f(\mathbf{x}_t)\|_1\right]\leq\mathcal{O}\left(\frac{d^{1/6}}{T^{1/3}}\right).$$

Note that the batch size for each iteration is $B_1=\mathcal{O}(d)$, by assuming that $N=B_1*T$, we know that the convergence concerning $N$ is

$$\mathcal{O}\left(\frac{d^{1/6}}{T^{1/3}}\right)=\mathcal{O}\left(\frac{d^{1/2}}{(B_1T)^{1/3}}\right)=\mathcal{O}\left(\frac{d^{1/2}}{N^{1/3}}\right).$$

Similar results can be easily obtained for $\alpha=1$, i.e., we can also guarantee the following for $\alpha=1$:

$$\mathbb{E}\left[\frac{1}{T}\sum_{t=1}^{T}\|\nabla f(\mathbf{x}_t)\|_1\right]\leq\mathcal{O}\left(\frac{d^{1/6}}{N^{1/3}}\right).$$

### E.1.1 Proof of Lemma 3

We prove this lemma by mathematical induction.

1) When $r=0$, we have the following:

$$\mathbb{E}\left[\left\|\sum_{s=1}^{t}(1-\beta)^{t-s}\Delta_s\right\|\right]=\mathbb{E}\left[\sqrt{\left\|\sum_{s=1}^{t}(1-\beta)^{t-s}\Delta_s\right\|^2}\right],$$

which satisfies the above lemma.

2) Then, suppose the lemma holds for $r = k$. For $r = k + 1$, we have

$$\mathbb{E}\left[\left\|\sum_{s=1}^{t}(1-\beta)^{t-s}\Delta_s\right\|\right]$$

$$\leq \mathbb{E}\left[\sqrt{\left\|\sum_{s=1}^{t-k}(1-\beta)^{t-s}\Delta_s\right\|^2 + \frac{1}{B_1}\sum_{s=1}^{k}(1-\beta)^{2s-2}\Lambda^2}\right] + \sum_{s=t+1-k}^{t}\frac{\Gamma}{\sqrt{B_1}}(1-\beta)^{t-s}\mathbb{E}\left[\|\nabla f(\mathbf{x}_s)\|\right]$$

$$= \mathbb{E}\left[\mathbb{E}_{\xi_{t-k}}\left[\sqrt{\left\|\sum_{s=1}^{t-k-1}(1-\beta)^{t-s}\Delta_s + (1-\beta)^k\Delta_{t-k}\right\|^2 + \frac{1}{B_1}\sum_{s=1}^{k}(1-\beta)^{2s-2}\Lambda^2}\right]\right]$$

$$+ \sum_{s=t+1-k}^{t}\frac{\Gamma}{\sqrt{B_1}}(1-\beta)^{t-s}\mathbb{E}\left[\|\nabla f(\mathbf{x}_s)\|\right]$$

$$\leq \mathbb{E}\left[\sqrt{\mathbb{E}_{\xi_{t-k}}\left\|\sum_{s=1}^{t-k-1}(1-\beta)^{t-s}\Delta_s + (1-\beta)^k\Delta_{t-k}\right\|^2 + \frac{1}{B_1}\sum_{s=1}^{k}(1-\beta)^{2s-2}\Lambda^2}\right]$$

$$+ \sum_{s=t+1-k}^{t}\frac{\Gamma}{\sqrt{B_1}}(1-\beta)^{t-s}\mathbb{E}\left[\|\nabla f(\mathbf{x}_s)\|\right]$$

$$\leq \mathbb{E}\left[\sqrt{\left\|\sum_{s=1}^{t-k-1}(1-\beta)^{t-s}\Delta_s\right\|^2 + \mathbb{E}_{\xi_{t-k}}\left[(1-\beta)^{2k}\|\Delta_{t-k}\|^2\right] + \frac{1}{B_1}\sum_{s=1}^{k}(1-\beta)^{2s-2}\Lambda^2}\right]$$

$$+ \sum_{s=t+1-k}^{t}\frac{\Gamma}{\sqrt{B_1}}(1-\beta)^{t-s}\mathbb{E}\left[\|\nabla f(\mathbf{x}_s)\|\right]$$

$$\leq \mathbb{E}\left[\sqrt{\left\|\sum_{s=1}^{t-k-1}(1-\beta)^{t-s}\Delta_s\right\|^2 + \frac{(1-\beta)^{2k}}{B_1}\left(\Lambda^2 + \Gamma^2\|\nabla f(\mathbf{x}_{t-k})\|^2\right) + \frac{1}{B_1}\sum_{s=1}^{k}(1-\beta)^{2s-2}\Lambda^2}\right]$$

$$+ \sum_{s=t+1-k}^{t}\frac{\Gamma}{\sqrt{B_1}}(1-\beta)^{t-s}\mathbb{E}\left[\|\nabla f(\mathbf{x}_s)\|\right]$$

$$\leq \mathbb{E}\left[\sqrt{\left\|\sum_{s=1}^{t-k-1}(1-\beta)^{t-s}\Delta_s\right\|^2 + \frac{1}{B_1}\sum_{s=1}^{k+1}(1-\beta)^{2s-2}\Lambda^2} + \frac{(1-\beta)^k\Gamma}{\sqrt{B_1}}\mathbb{E}\left[\|\nabla f(\mathbf{x}_{t-k})\|\right]\right]$$

$$+ \sum_{s=t+1-k}^{t}\frac{\Gamma}{\sqrt{B_1}}(1-\beta)^{t-s}\mathbb{E}\left[\|\nabla f(\mathbf{x}_s)\|\right]$$

$$= \mathbb{E}\left[\sqrt{\left\|\sum_{s=1}^{t-k-1}(1-\beta)^{t-s}\Delta_s\right\|^2 + \frac{1}{B_1}\sum_{s=1}^{k+1}(1-\beta)^{2s-2}\Lambda^2}\right] + \sum_{s=t-k}^{t}\frac{\Gamma}{\sqrt{B_1}}(1-\beta)^{t-s}\mathbb{E}\left[\|\nabla f(\mathbf{x}_s)\|\right],$$

where the second inequality is due to the Jensen Inequality.

### E.1.2 Proof of Lemma 4

We prove this lemma by mathematical induction.

1) When $r = 0$, we can easily prove $\mathbb{E}\left[\left\|\sum_{s=1}^{t}(1-\beta)^{t-s}G_s\right\|\right] = \mathbb{E}\left[\sqrt{\left\|\sum_{s=1}^{t}(1-\beta)^{t-s}G_s\right\|^2}\right]$

2) Suppose the inequality holds for $r = k$. Then, for $r = k + 1$, we derive

$$\mathbb{E}\left[\left\|\sum_{s=1}^{t}(1-\beta)^{t-s}G_s\right\|\right]$$

$$\leq \mathbb{E}\left[\sqrt{\left\|\sum_{s=1}^{t-k}(1-\beta)^{t-s}G_s\right\|^2 + \frac{1}{B_1}\sum_{s=1}^{k}2(1-\beta)^{2s-2}\eta^2 L_3^2 d}\right]$$

$$+ \frac{\sqrt{2d}\eta L_4}{\sqrt{B_1}}\sum_{s=t+1-k}^{t}(1-\beta)^{t-s}\mathbb{E}\left[\|\nabla f(\mathbf{x}_s)\|\right]$$

$$= \mathbb{E}\left[\mathbb{E}_{\xi_{t-k}}\left[\sqrt{\left\|\sum_{s=1}^{t-k}(1-\beta)^{t-s}G_s\right\|^2 + \frac{1}{B_1}\sum_{s=1}^{k}2(1-\beta)^{2s-2}\eta^2 L_3^2 d}\right]\right]$$

$$+ \frac{\sqrt{2d}\eta L_4}{\sqrt{B_1}}\sum_{s=t+1-k}^{t}(1-\beta)^{t-s}\mathbb{E}\left[\|\nabla f(\mathbf{x}_s)\|\right]$$

$$= \mathbb{E}\left[\mathbb{E}_{\xi_{t-k}}\left[\sqrt{\left\|\sum_{s=1}^{t-k-1}(1-\beta)^{t-s}G_s + (1-\beta)^k G_{t-k}\right\|^2 + \frac{1}{B_1}\sum_{s=1}^{k}2(1-\beta)^{2s-2}\eta^2 L_3^2 d}\right]\right]$$

$$+ \frac{\sqrt{2d}\eta L_4}{\sqrt{B_1}}\sum_{s=t+1-k}^{t}(1-\beta)^{t-s}\mathbb{E}\left[\|\nabla f(\mathbf{x}_s)\|\right]$$

$$\leq \mathbb{E}\left[\sqrt{\mathbb{E}_{\xi_{t-k}}\left\|\sum_{s=1}^{t-k-1}(1-\beta)^{t-s}G_s + (1-\beta)^k G_{t-k}\right\|^2 + \frac{1}{B_1}\sum_{s=1}^{k}2(1-\beta)^{2s-2}\eta^2 L_3^2 d}\right]$$

$$+ \frac{\sqrt{2d}\eta L_4}{\sqrt{B_1}}\sum_{s=t+1-k}^{t}(1-\beta)^{t-s}\mathbb{E}\left[\|\nabla f(\mathbf{x}_s)\|\right]$$

$$= \mathbb{E}\left[\sqrt{\left\|\sum_{s=1}^{t-k-1}(1-\beta)^{t-s}G_s\right\|^2 + (1-\beta)^{2k}\mathbb{E}_{\xi_{t-k}}\left[\|G_{t-k}\|^2\right] + \frac{1}{B_1}\sum_{s=1}^{k}2(1-\beta)^{2s-2}\eta^2 L_3^2 d}\right]$$

$$+ \frac{\sqrt{2d}\eta L_4}{\sqrt{B_1}}\sum_{s=t+1-k}^{t}(1-\beta)^{t-s}\mathbb{E}\left[\|\nabla f(\mathbf{x}_s)\|\right]$$

$$\leq \mathbb{E}\left[\sqrt{\left\|\sum_{s=1}^{t-k-1}(1-\beta)^{t-s}G_s\right\|^2 + \frac{2}{B_1}(1-\beta)^{2k}\eta^2 L_3^2 d + \frac{1}{B_1}\sum_{s=1}^{k}2(1-\beta)^{2s-2}\eta^2 L_3^2 d}\right]$$

$$+ \frac{\sqrt{2d}\eta L_4(1-\beta)^k}{\sqrt{B_1}}\mathbb{E}\left[\|\nabla f(\mathbf{x}_{t-k})\|\right] + \frac{\sqrt{2d}\eta L_4}{\sqrt{B_1}}\sum_{s=t+1-k}^{t}(1-\beta)^{t-s}\mathbb{E}\left[\|\nabla f(\mathbf{x}_s)\|\right]$$

$$= \mathbb{E}\left[\sqrt{\left\|\sum_{s=1}^{t-k-1}(1-\beta)^{t-s}G_s\right\|^2 + \frac{1}{B_1}\sum_{s=1}^{k+1}2(1-\beta)^{2s-2}\eta^2 L_3^2 d}\right]$$

$$+ \frac{\sqrt{2d}\eta L_4}{\sqrt{B_1}}\sum_{s=t-k}^{t}(1-\beta)^{t-s}\mathbb{E}\left[\|\nabla f(\mathbf{x}_s)\|\right],$$

where the third inequality is due to the following, for simplify we denote $\xi_{t-k} = \xi_{t-k}^1, ..., \xi_{t-k}^{B_1}$:

$$\mathbb{E}_{\xi_{t-k}}\left[\|G_{t-k}\|^2\right] = \mathbb{E}_{\xi_{t-k}}\left[\frac{1}{B_1^2}\sum_{j=1}^{B_1}\left\|\nabla f(\mathbf{x}_{t-k};\xi_{t-k}^j) - \nabla f(\mathbf{x}_{t-k-1};\xi_{t-k}^j) + \nabla f(\mathbf{x}_{t-k-1}) - \nabla f(\mathbf{x}_{t-k})\right\|^2\right]$$

$$= \mathbb{E}_{\xi_{t-k}}\left[\frac{1}{B_1^2}\sum_{j=1}^{B_1}\left\|\nabla f(\mathbf{x}_{t-k};\xi_{t-k}^j) - \nabla f(\mathbf{x}_{t-k-1};\xi_{t-k}^j)\right\|^2\right] + \frac{1}{B_1}\left[\|\nabla f(\mathbf{x}_{t-k-1}) - \nabla f(\mathbf{x}_{t-k})\|^2\right]$$

$$- 2\mathbb{E}_{\xi_{t-k}}\left[\frac{1}{B_1^2}\sum_{j=1}^{B_1}\left\langle\nabla f(\mathbf{x}_{t-k};\xi_{t-k}^j) - \nabla f(\mathbf{x}_{t-k-1};\xi_{t-k}^j), \nabla f(\mathbf{x}_{t-k}) - \nabla f(\mathbf{x}_{t-k-1})\right\rangle\right]$$

$$= \mathbb{E}_{\xi_{t-k}}\left[\frac{1}{B_1^2}\sum_{j=1}^{B_1}\left\|\nabla f(\mathbf{x}_{t-k};\xi_{t-k}^j) - \nabla f(\mathbf{x}_{t-k-1};\xi_{t-k}^j)\right\|^2\right] - \frac{1}{B_1}\left[\|\nabla f(\mathbf{x}_{t-k-1}) - \nabla f(\mathbf{x}_{t-k})\|^2\right]$$

$$\leq \mathbb{E}_{\xi_{t-k}}\left[\frac{1}{B_1^2}\sum_{j=1}^{B_1}\left\|\nabla f(\mathbf{x}_{t-k};\xi_{t-k}^j) - \nabla f(\mathbf{x}_{t-k-1};\xi_{t-k}^j)\right\|^2\right]$$

For $\alpha \in (0,1)$, denoting $L_3^2 = \left(\overline{K}_0 + \overline{K}_1 + \overline{K}_2\right)^2 + \overline{K}_1^2\Lambda^2$, and $L_4^2 = \overline{K}_1^2(1 + \Gamma^2)$, we have:

$$\mathbb{E}_{\xi_{t-k}}\left[\frac{1}{B_1^2}\sum_{j=1}^{B_1}\left\|\nabla f(\mathbf{x}_{t-k};\xi_{t-k}^j) - \nabla f(\mathbf{x}_{t-k-1};\xi_{t-k}^j)\right\|^2\right]$$

$$\leq \frac{1}{B_1^2}\sum_{j=1}^{B_1}\|\mathbf{x}_{t-k} - \mathbf{x}_{t-k-1}\|^2\left(\overline{K}_0 + \overline{K}_1\mathbb{E}_{\xi_{t-k}}\left[\left\|\nabla f(\mathbf{x}_{t-k};\xi_{t-k}^j)\right\|^\alpha\right] + \overline{K}_2\|\mathbf{x}_{t-k} - \mathbf{x}_{t-k-1}\|^{\frac{\alpha}{1-\alpha}}\right)^2$$

$$\leq \frac{\eta^2 d}{B_1^2}\sum_{j=1}^{B_1}\left(\overline{K}_0 + \overline{K}_1 + \overline{K}_2 + \overline{K}_1\mathbb{E}_{\xi_{t-k}}\left[\left\|\nabla f(\mathbf{x}_{t-k};\xi_{t-k}^j)\right\|\right]\right)^2$$

$$\leq \frac{2\eta^2 d}{B_1}\left(\overline{K}_0 + \overline{K}_1 + \overline{K}_2\right)^2 + \frac{2\eta^2 d}{B_1^2}\sum_{j=1}^{B_1}\overline{K}_1^2\left(\mathbb{E}_{\xi_{t-k}}\left[\left\|\nabla f(\mathbf{x}_{t-k};\xi_{t-k}^j)\right\|\right]\right)^2$$

$$\leq \frac{2\eta^2 d}{B_1}\left(\overline{K}_0 + \overline{K}_1 + \overline{K}_2\right)^2 + \frac{2\eta^2 d}{B_1}\overline{K}_1^2\left((1 + \Gamma^2)\|\nabla f(\mathbf{x}_{t-k})\| + \Lambda^2\right)$$

$$\leq \frac{2\eta^2 d L_3^2}{B_1} + \frac{2\eta^2 d L_4^2}{B_1}\|\nabla f(\mathbf{x}_{t-k})\|^2,$$

where the second inequality holds by setting $\eta \leq d^{-1/2}$ such that $\|\mathbf{x}_{t-k} - \mathbf{x}_{t-k-1}\| \leq \eta\sqrt{d} \leq 1$.

For $\alpha = 1$, denoting $L_3^2 = 3\left(L_0^2 + 2L_1^2\Lambda^2\right)$, and $L_4^2 = 6L_1^2(1 + \Gamma^2)$, we have:

$$\mathbb{E}_{\xi_{t-k}}\left[\frac{1}{B_1^2}\sum_{j=1}^{B_1}\left\|\nabla f(\mathbf{x}_{t-k};\xi_{t-k}^j) - \nabla f(\mathbf{x}_{t-k-1};\xi_{t-k}^j)\right\|^2\right]$$

$$\leq \frac{2}{B_1^2}\sum_{j=1}^{B_1}\|\mathbf{x}_{t-k} - \mathbf{x}_{t-k-1}\|^2\left(L_0^2 + 2L_1^2\mathbb{E}_{\xi_{t-k}}\left[\left\|\nabla f(\mathbf{x}_{t-k};\xi_{t-k}^j)\right\|^2\right]\right)\exp\left(12L_1^2\|\mathbf{x}_{t-k} - \mathbf{x}_{t-k-1}\|^2\right)$$

$$\leq \frac{6\eta^2 d}{B_1^2}\sum_{j=1}^{B_1}\left(L_0^2 + 2L_1^2\mathbb{E}_{\xi_{t-k}}\left[\left\|\nabla f(\mathbf{x}_{t-k};\xi_{t-k}^j)\right\|^2\right]\right)$$

$$\leq \frac{6\eta^2 d}{B_1}\left(L_0^2 + 2L_1^2\left((1 + \Gamma^2)\|\nabla f(\mathbf{x}_{t-k})\|^2 + \Lambda^2\right)\right)$$

$$\leq \frac{2\eta^2 d L_3^2}{B_1} + \frac{2\eta^2 d L_4^2}{B_1}\|\nabla f(\mathbf{x}_{t-k})\|^2,$$

where the second inequality holds by setting $\eta \leq \frac{1}{\sqrt{12L_1^2 d}}$, such that $\|\mathbf{x}_{t-k} - \mathbf{x}_{t-k-1}\|^2 \leq \frac{1}{12L_1^2}$.

## E.2   Proof of Theorem 6

Similar to Lemma 1 and 2, we can have the following lemma for generalized individual smoothness.

**Lemma 5** *For $\alpha \in (0,1)$, generalized individual smoothness (Assumption 3') leads to*

$$f_i(\mathbf{x}_{t+1}) \leq f_i(\mathbf{x}_t) + \langle \nabla f_i(\mathbf{x}_t), \mathbf{x}_{t+1} - \mathbf{x}_t \rangle$$
$$+ \frac{1}{2}\|\mathbf{x}_{t+1} - \mathbf{x}_t\|^2 \left( K_0 + K_1\|\nabla f(\mathbf{x}_t)\|^\alpha + 2K_2\|\mathbf{x}_{t+1} - \mathbf{x}_t\|^{\frac{\alpha}{1-\alpha}} \right),$$

*as well as*

$$\|\nabla f_i(\mathbf{x}_{t+1}) - \nabla f_i(\mathbf{x}_t)\| \leq \|\mathbf{x}_{t+1} - \mathbf{x}_t\| \left( K_0 + K_1\|\nabla f(\mathbf{x}_t)\|^\alpha + K_2\|\mathbf{x}_{t+1} - \mathbf{x}_t\|^{\frac{\alpha}{1-\alpha}} \right).$$

*where $K_0 := L_0\left(2^{\frac{\alpha^2}{1-\alpha}} + 1\right)$, $K_1 := L_1 \cdot 2^{\frac{\alpha^2}{1-\alpha}} \cdot 3^\alpha$, $K_2 := L_1^{\frac{1}{1-\alpha}} \cdot 2^{\frac{\alpha^2}{1-\alpha}} \cdot 3^\alpha (1-\alpha)^{\frac{\alpha}{1-\alpha}}$.*

**Lemma 6** *For $\alpha = 1$, generalized individual smoothness (Assumption 3') leads to*

$$f_i(\mathbf{x}_{t+1}) \leq f_i(\mathbf{x}_t) + \langle \nabla f_i(\mathbf{x}_t), \mathbf{x}_{t+1} - \mathbf{x}_t \rangle$$
$$+ \frac{1}{2}\|\mathbf{x}_{t+1} - \mathbf{x}_t\|^2 \left( L_0 + L_1\|\nabla f(\mathbf{x}_t)\| \right) \exp\left( L_1\|\mathbf{x}_{t+1} - \mathbf{x}_t\| \right),$$

*as well as*

$$\|\nabla f_i(\mathbf{x}_{t+1}) - \nabla f_i(\mathbf{x}_t)\| \leq \|\mathbf{x}_{t+1} - \mathbf{x}_t\| \left( L_0 + L_1\|\nabla f(\mathbf{x}_t)\| \right) \exp\left( L_1\|\mathbf{x}_{t+1} - \mathbf{x}_t\| \right).$$

Then, we can begin our proof. For $\alpha \in (0,1)$, according to Lemma 5, by setting $\eta \leq d^{-\frac{1}{2}}$, we have:

$$f(\mathbf{x}_{t+1}) - f(\mathbf{x}_t)$$
$$\leq \frac{1}{m}\sum_{i=1}^m f_i(\mathbf{x}_{t+1}) - \frac{1}{m}\sum_{i=1}^m f_i(\mathbf{x}_t)$$
$$\leq \frac{1}{m}\sum_{i=1}^m \langle \nabla f_i(\mathbf{x}_t), \mathbf{x}_{t+1} - \mathbf{x}_t \rangle + \frac{1}{2}\|\mathbf{x}_{t+1} - \mathbf{x}_t\|^2 \left( K_0 + K_1\|\nabla f(\mathbf{x}_t)\|^\alpha + 2K_2\|\mathbf{x}_{t+1} - \mathbf{x}_t\|^{\frac{\alpha}{1-\alpha}} \right)$$
$$\leq \langle \nabla f(\mathbf{x}_t), -\eta\,\mathrm{Sign}(\mathbf{v}_t) \rangle + \frac{1}{2}\|\mathbf{x}_{t+1} - \mathbf{x}_t\|^2 \left( K_0 + K_1\left(1 + \|\nabla f(\mathbf{x}_t)\|\right) + 2K_2 \right)$$
$$\leq \eta\langle \nabla f(\mathbf{x}_t), \mathrm{Sign}(\nabla f(\mathbf{x}_t)) - \mathrm{Sign}(\mathbf{v}_t) \rangle - \eta\langle \nabla f(\mathbf{x}_t), \mathrm{Sign}(\nabla f(\mathbf{x}_t)) \rangle$$
$$+ \frac{1}{2}\|\mathbf{x}_{t+1} - \mathbf{x}_t\|^2 \left( K_0 + K_1\left(1 + \|\nabla f(\mathbf{x}_t)\|\right) + 2K_2 \right)$$
$$= \eta\langle \nabla f(\mathbf{x}_t), \mathrm{Sign}(\nabla f(\mathbf{x}_t)) - \mathrm{Sign}(\mathbf{v}_t) \rangle - \eta\|\nabla f(\mathbf{x}_t)\|_1$$
$$+ \frac{1}{2}\|\mathbf{x}_{t+1} - \mathbf{x}_t\|^2 \left( (K_0 + K_1 + 2K_2) + K_1\|\nabla f(\mathbf{x}_t)\| \right)$$
$$\leq 2\eta\sqrt{d}\|\nabla f(\mathbf{x}_t) - \mathbf{v}_t\| - \eta\|\nabla f(\mathbf{x}_t)\|_1 + \frac{\eta^2 d}{2}\left( (K_0 + K_1 + 2K_2) + K_1\|\nabla f(\mathbf{x}_t)\| \right),$$

where the second inequality is due to the fact that $\alpha < 1$ and $\|\mathbf{x}_{t+1} - \mathbf{x}_t\|^2 \leq \eta^2 d \leq 1$.

Rearranging and summing up, we then have

$$\mathbb{E}\left[ \frac{1}{T}\sum_{t=1}^T \|\nabla f(\mathbf{x}_t)\|_1 \right] \leq \frac{\Delta_f}{\eta T} + 2\sqrt{d} \cdot \mathbb{E}\left[ \frac{1}{T}\sum_{t=1}^T \|\nabla f(\mathbf{x}_t) - \mathbf{v}_t\| \right]$$
$$+ \frac{\eta d(K_0 + K_1 + 2K_2)}{2} + \frac{\eta d K_1}{2T}\mathbb{E}\left[ \sum_{t=1}^T \|\nabla f(\mathbf{x}_t)\| \right].$$

By setting $\eta \leq \min\{\frac{1}{\sqrt{d}}, \frac{1}{dK_1}\}$, we can get

$$\mathbb{E}\left[ \frac{1}{T}\sum_{t=1}^T \|\nabla f(\mathbf{x}_t)\|_1 \right] \leq \frac{2\Delta_f}{\eta T} + 4\sqrt{d} \cdot \mathbb{E}\left[ \frac{1}{T}\sum_{t=1}^T \|\nabla f(\mathbf{x}_t) - \mathbf{v}_t\| \right] + \eta d(K_0 + K_1 + 2K_2).$$
$$\tag{17}$$

For $\alpha = 1$, by setting $\eta \leq \frac{1}{3dL_1}$, we can apply the very similar analysis and obtain

$$\mathbb{E}\left[\frac{1}{T}\sum_{t=1}^{T}\|\nabla f(\mathbf{x}_t)\|_1\right] \leq \frac{2\Delta_f}{\eta T} + 4\sqrt{d}\cdot\mathbb{E}\left[\frac{1}{T}\sum_{t=1}^{T}\|\nabla f(\mathbf{x}_t) - \mathbf{v}_t\|\right] + 3\eta dL_0. \qquad (18)$$

Then we bound the term $\mathbb{E}\left[\frac{1}{T}\sum_{t=1}^{T}\|\nabla f(\mathbf{x}_t) - \mathbf{v}_t\|\right]$. According to the definition of $\mathbf{v}_t$, we have:

$$\begin{aligned}\mathbf{v}_t - \nabla f(\mathbf{x}_t) =&(1-\beta)\left(\mathbf{v}_{t-1} - \nabla f(\mathbf{x}_{t-1})\right) + \beta\left(\mathbf{h}_t - \nabla f(\mathbf{x}_t)\right) \\ &+ (1-\beta)\left(\nabla f(\mathbf{x}_t;\xi_t^k) - \nabla f(\mathbf{x}_{t-1};\xi_t^k) + \nabla f(\mathbf{x}_{t-1}) - \nabla f(\mathbf{x}_t)\right)\end{aligned}$$

Denote that $G_t = \left(\nabla f(\mathbf{x}_t;\xi_t^k) - \nabla f(\mathbf{x}_{t-1};\xi_t^k) + \nabla f(\mathbf{x}_{t-1}) - \nabla f(\mathbf{x}_t)\right)$ and $\Delta_t = \mathbf{h}_t - \nabla f(\mathbf{x}_t)$. By summing up, we have

$$\begin{aligned}&\mathbf{v}_t - \nabla f(\mathbf{x}_t) \\ =&(1-\beta)(\mathbf{v}_{t-1} - \nabla f(\mathbf{x}_{t-1})) + \beta\Delta_t + (1-\beta)G_t \\ =&(1-\beta)^{t-1}\left(\mathbf{v}_1 - \nabla f(\mathbf{x}_1)\right) + \beta\sum_{s=1}^{t}(1-\beta)^{t-s}\Delta_s + (1-\beta)\sum_{s=1}^{t}(1-\beta)^{t-s}G_s.\end{aligned}$$

Thus, we can know that

$$\begin{aligned}&\mathbb{E}\left[\|\mathbf{v}_t - \nabla f(\mathbf{x}_t)\|\right] \\ \leq&(1-\beta)^{t-1}\mathbb{E}\left[\|\mathbf{v}_1 - \nabla f(\mathbf{x}_1)\|\right] + \beta\mathbb{E}\left[\left\|\sum_{s=1}^{t}(1-\beta)^{t-s}\Delta_s\right\|\right] + (1-\beta)\mathbb{E}\left[\left\|\sum_{s=1}^{t}(1-\beta)^{t-s}G_s\right\|\right] \\ \leq&\beta\mathbb{E}\left[\left\|\sum_{s=1}^{t}(1-\beta)^{t-s}\Delta_s\right\|\right] + (1-\beta)\mathbb{E}\left[\left\|\sum_{s=1}^{t}(1-\beta)^{t-s}G_s\right\|\right],\end{aligned}$$

where the first term vanishes since we use full batch in the first iteration.

Then we give the following two important lemmas, and their proofs can be found in Appendix E.2.1 and Appendix E.2.2, respectively.

**Lemma 7**

$$\mathbb{E}\left[\left\|\sum_{s=1}^{t}(1-\beta)^{t-s}\Delta_s\right\|\right] \leq \mathbb{E}\left[\sqrt{\left\|\sum_{s=1}^{t-r}(1-\beta)^{t-s}\Delta_s\right\|^2 + \sum_{s=1}^{r}2(1-\beta)^{2s-2}\eta^2 I^2 L_5^2 d}\right]$$

$$+ \sqrt{2d}\eta I L_6 \sum_{s=t+1-r}^{t}(1-\beta)^{t-s}\mathbb{E}\left[\|\nabla f(\mathbf{x}_s)\|\right]$$

**Lemma 8**

$$\mathbb{E}\left[\left\|\sum_{s=1}^{t}(1-\beta)^{t-s}G_s\right\|\right] \leq \mathbb{E}\left[\sqrt{\left\|\sum_{s=1}^{t-r}(1-\beta)^{t-s}G_s\right\|^2 + \sum_{s=1}^{r}2(1-\beta)^{2s-2}\eta^2 L_7^2 d}\right]$$

$$+ \sqrt{2d}\eta L_8 \sum_{s=t+1-r}^{t}(1-\beta)^{t-s}\mathbb{E}\left[\|\nabla f(\mathbf{x}_s)\|\right]$$

Using these lemmas and setting $r = t$, we then have

$$\mathbb{E}\left[\left\|\sum_{s=1}^{t}(1-\beta)^{t-s}\Delta_s\right\|\right]$$

$$\leq\sqrt{2\eta^2 I^2 L_5^2 d\sum_{s=1}^{t}(1-\beta)^{2s-2} + \sqrt{2d}\eta I L_6\sum_{s=1}^{t}(1-\beta)^{t-s}\mathbb{E}\left[\|\nabla f(\mathbf{x}_s)\|\right]}$$

$$\leq\frac{\sqrt{2d}\eta I L_5}{\sqrt{\beta}} + \sqrt{2d}\eta I L_6\sum_{s=1}^{t}(1-\beta)^{t-s}\mathbb{E}\left[\|\nabla f(\mathbf{x}_s)\|\right],$$

as well as

$$\mathbb{E}\left[\left\|\sum_{s=1}^{t}(1-\beta)^{t-s}G_s\right\|\right]$$

$$\leq\sqrt{2\eta^2 L_7^2 d\sum_{s=1}^{t}(1-\beta)^{2s-2} + \sqrt{2d}\eta L_8\sum_{s=1}^{t}(1-\beta)^{t-s}\mathbb{E}\left[\|\nabla f(\mathbf{x}_s)\|\right]}$$

$$\leq\frac{\sqrt{2d}\eta L_7}{\sqrt{\beta}} + \sqrt{2d}\eta L_8\sum_{s=1}^{t}(1-\beta)^{t-s}\mathbb{E}\left[\|\nabla f(\mathbf{x}_s)\|\right].$$

Combining above inequalities and setting $\beta = 1/m$ and $I = m$, we derive

$$\mathbb{E}\left[\frac{1}{T}\sum_{t=1}^{T}\|\nabla f(\mathbf{x}_t) - \mathbf{v}_t\|\right]$$

$$\leq\beta\frac{1}{T}\sum_{t=1}^{T}\mathbb{E}\left[\left\|\sum_{s=1}^{t}(1-\beta)^{t-s}\Delta_s\right\|\right] + \frac{1}{T}\sum_{t=1}^{T}\mathbb{E}\left[\left\|\sum_{s=1}^{t}(1-\beta)^{t-s}G_s\right\|\right]$$

$$\leq\sqrt{2d\beta}\eta I L_5 + \frac{\sqrt{2d}\eta L_7}{\sqrt{\beta}} + \left(\sqrt{2d}\eta\beta I L_6 + \sqrt{2d}\eta L_8\right)\frac{1}{T}\sum_{t=1}^{T}\sum_{s=1}^{t}(1-\beta)^{t-s}\mathbb{E}\left[\|\nabla f(\mathbf{x}_s)\|\right]$$

$$\leq\sqrt{2dm}\eta(L_5 + L_7) + \sqrt{2d}\eta(L_6 + L_8)\left(\sum_{i=1}^{T}(1-\beta)^i\right)\frac{1}{T}\sum_{t=1}^{T}\mathbb{E}\left[\|\nabla f(\mathbf{x}_t)\|\right]$$

$$\leq\sqrt{2dm}\eta(L_5 + L_7) + \sqrt{2d}\eta m(L_6 + L_8)\frac{1}{T}\sum_{t=1}^{T}\mathbb{E}\left[\|\nabla f(\mathbf{x}_t)\|\right].$$

For $\alpha \in (0,1)$, by setting $\eta = \min\left\{\frac{1}{m^{1/4}d^{1/2}T^{1/2}}, \frac{1}{8\sqrt{2}md(L_6+L_8+1)}\right\}$, we can guarantee

$$\mathbb{E}\left[\frac{1}{T}\sum_{t=1}^{T}\|\nabla f(\mathbf{x}_t)\|_1\right]$$

$$\leq\frac{2\Delta_f}{\eta T} + \eta d(K_0 + K_1 + 2K_2) + 4d\sqrt{2m}\eta(L_5 + L_7) + 4d\sqrt{2}\eta m(L_6 + L_8)\frac{1}{T}\sum_{t=1}^{T}\mathbb{E}\left[\|\nabla f(\mathbf{x}_t)\|\right]$$

$$\leq\mathcal{O}\left(\frac{m^{1/4}d^{1/2}}{T^{1/2}} + \frac{md}{T}\right) + \frac{1}{2}\mathbb{E}\left[\frac{1}{T}\sum_{t=1}^{T}\|\nabla f(\mathbf{x}_t)\|_1\right],$$

which indicates that $\mathbb{E}\left[\frac{1}{T}\sum_{t=1}^{T}\|\nabla f(\mathbf{x}_t)\|_1\right] \leq \mathcal{O}\left(\frac{m^{1/4}d^{1/2}}{T^{1/2}} + \frac{md}{T}\right)$. Similar results can be easily obtained for $\alpha = 1$.

### E.2.1 Proof of Lemma 7

We prove this lemma by mathematical induction.

1) When $r = 0$, we have the following:

$$\mathbb{E}\left[\left\|\sum_{s=1}^{t}(1-\beta)^{t-s}\Delta_s\right\|\right] = \mathbb{E}\left[\sqrt{\left\|\sum_{s=1}^{t}(1-\beta)^{t-s}\Delta_s\right\|^2}\right],$$

which satisfies the above lemma.

2) Then, suppose the lemma holds for $r = k$. For $r = k+1$, we have

$$\mathbb{E}\left[\left\|\sum_{s=1}^{t}(1-\beta)^{t-s}\Delta_s\right\|\right]$$

$$\leq \mathbb{E}\left[\sqrt{\left\|\sum_{s=1}^{t-k}(1-\beta)^{t-s}\Delta_s\right\|^2 + \sum_{s=1}^{k}2(1-\beta)^{2s-2}\eta^2 I^2 L_5^2 d}\right]$$

$$+ \sqrt{2d}\eta I L_6 \sum_{s=t+1-k}^{t}(1-\beta)^{t-s}\mathbb{E}\left[\|\nabla f(\mathbf{x}_s)\|\right]$$

$$= \mathbb{E}\left[\mathbb{E}_{i_{t-k}}\left[\sqrt{\left\|\sum_{s=1}^{t-k-1}(1-\beta)^{t-s}\Delta_s + (1-\beta)^k\Delta_{t-k}\right\|^2 + \sum_{s=1}^{k}2(1-\beta)^{2s-2}\eta^2 I^2 L_5^2 d}\right]\right]$$

$$+ \sqrt{2d}\eta I L_6 \sum_{s=t+1-k}^{t}(1-\beta)^{t-s}\mathbb{E}\left[\|\nabla f(\mathbf{x}_s)\|\right]$$

$$\leq \mathbb{E}\left[\sqrt{\mathbb{E}_{i_{t-k}}\left\|\sum_{s=1}^{t-k-1}(1-\beta)^{t-s}\Delta_s + (1-\beta)^k\Delta_{t-k}\right\|^2 + \sum_{s=1}^{k}2(1-\beta)^{2s-2}\eta^2 I^2 L_5^2 d}\right]$$

$$+ \sqrt{2d}\eta I L_6 \sum_{s=t+1-k}^{t}(1-\beta)^{t-s}\mathbb{E}\left[\|\nabla f(\mathbf{x}_s)\|\right]$$

$$= \mathbb{E}\left[\sqrt{\left\|\sum_{s=1}^{t-k-1}(1-\beta)^{t-s}\Delta_s\right\|^2 + (1-\beta)^{2k}\mathbb{E}_{i_{t-k}}\left[\|\Delta_{t-k}\|^2\right] + \sum_{s=1}^{k}2(1-\beta)^{2s-2}\eta^2 I^2 L_5^2 d}\right]$$

$$+ \sqrt{2d}\eta I L_6 \sum_{s=t+1-k}^{t}(1-\beta)^{t-s}\mathbb{E}\left[\|\nabla f(\mathbf{x}_s)\|\right]$$

$$\leq \mathbb{E}\left[\sqrt{\left\|\sum_{s=1}^{t-k-1}(1-\beta)^{t-s}\Delta_s\right\|^2 + 2(1-\beta)^{2k}\eta^2 I^2 L_5^2 d + \sum_{s=1}^{k}2(1-\beta)^{2s-2}\eta^2 I^2 L_5^2 d}\right]$$

$$+ \sqrt{2d}\eta I L_6 (1-\beta)^k \mathbb{E}\left[\|\nabla f(\mathbf{x}_{t-k})\|\right] + \sqrt{2d}\eta I L_6 \sum_{s=t+1-k}^{t}(1-\beta)^{t-s}\mathbb{E}\left[\|\nabla f(\mathbf{x}_s)\|\right]$$

$$= \mathbb{E}\left[\sqrt{\left\|\sum_{s=1}^{t-k-1}(1-\beta)^{t-s}\Delta_s\right\|^2 + \sum_{s=1}^{k+1}2(1-\beta)^{2s-2}\eta^2 I^2 L_5^2 d}\right]$$

$$+ \sqrt{2d}\eta I L_6 \sum_{s=t-k}^{t}(1-\beta)^{t-s}\mathbb{E}\left[\|\nabla f(\mathbf{x}_s)\|\right],$$

where the third inequality is due to the following:

$$\mathbb{E}_{i_{t-k}}\left[\|\Delta_{t-k}\|^2\right] = \mathbb{E}_{i_{t-k}}\left[\|\nabla f_{i_{t-k}}(\mathbf{x}_{t-k}) - \nabla f_{i_{t-k}}(\mathbf{x}_\tau) + \nabla f(\mathbf{x}_\tau) - \nabla f(\mathbf{x}_{t-k})\|^2\right]$$
$$\leq \mathbb{E}_{i_{t-k}}\left[\|\nabla f_{i_{t-k}}(\mathbf{x}_{t-k}) - \nabla f_{i_{t-k}}(\mathbf{x}_\tau)\|^2\right]$$

For $\alpha \in (0, 1)$, denoting that

$$L_5^2 = \left(K_0 + K_1 + K_2\right)^2,$$
$$L_6^2 = K_1^2,$$

we have:

$$\mathbb{E}_{i_{t-k}}\left[\|\nabla f_{i_{t-k}}(\mathbf{x}_{t-k}) - \nabla f_{i_{t-k}}(\mathbf{x}_\tau)\|^2\right]$$
$$\leq \|\mathbf{x}_{t-k} - \mathbf{x}_\tau\|^2 \left(K_0 + K_1 \|\nabla f(\mathbf{x}_{t-k})\|^\alpha + K_2\|\mathbf{x}_{t-k} - \mathbf{x}_\tau\|^{\frac{\alpha}{1-\alpha}}\right)^2$$
$$\leq \eta^2 I^2 d \left(K_0 + K_1 + K_2 + K_1 \|\nabla f(\mathbf{x}_{t-k})\|\right)^2$$
$$\leq 2\eta^2 I^2 d \left(K_0 + K_1 + K_2\right)^2 + 2\eta^2 I^2 d K_1^2 \|\nabla f(\mathbf{x}_{t-k})\|^2$$
$$\leq 2\eta^2 d I^2 L_5^2 + 2\eta^2 I^2 d L_6^2 \|\nabla f(\mathbf{x}_{t-k})\|^2,$$

where the second inequality holds by setting

$$\eta \leq I^{-1} d^{-1/2}$$

such that

$$\|\mathbf{x}_{t-k} - \mathbf{x}_\tau\| \leq \eta I \sqrt{d} \leq 1.$$

For $\alpha = 1$, denoting that

$$L_5^2 = 9L_0^2$$
$$L_6^2 = 9L_1^2,$$

we have:

$$\mathbb{E}_{i_{t-k}}\left[\|\nabla f_{i_{t-k}}(\mathbf{x}_{t-k}) - \nabla f_{i_{t-k}}(\mathbf{x}_\tau)\|^2\right]$$
$$\leq 2 \|\mathbf{x}_{t-k} - \mathbf{x}_\tau\|^2 \left(L_0^2 + L_1^2 \|\nabla f(\mathbf{x}_{t-k})\|^2\right) \left(\exp\left(L_1^2 \|\mathbf{x}_{t-k} - \mathbf{x}_\tau\|^2\right)\right)^2$$
$$\leq 18\eta^2 d I^2 \left(L_0^2 + L_1^2 \|\nabla f(\mathbf{x}_{t-k})\|^2\right)$$
$$\leq 2\eta^2 I^2 d L_5^2 + 2\eta^2 I^2 d L_6^2 \|\nabla f(\mathbf{x}_{t-k})\|^2,$$

where the second inequality holds by setting

$$\eta \leq \frac{1}{\sqrt{L_1^2 I^2 d}},$$

such that we have

$$\|\mathbf{x}_{t-k} - \mathbf{x}_{t-k-1}\|^2 \leq \eta^2 I^2 d \leq \frac{1}{L_1^2}.$$

### E.2.2   Proof of Lemma 8

We prove this lemma by mathematical induction.

1) When $r = 0$, we can easily prove that

$$\mathbb{E}\left[\left\|\sum_{s=1}^t (1-\beta)^{t-s} G_s\right\|\right] = \mathbb{E}\left[\sqrt{\left\|\sum_{s=1}^t (1-\beta)^{t-s} G_s\right\|^2}\right].$$

2) Suppose the inequality holds for $r = k$. Then, for $r = k + 1$, we derive

$$\mathbb{E}\left[\left\|\sum_{s=1}^{t}(1-\beta)^{t-s}G_s\right\|\right]$$

$$\leq \mathbb{E}\left[\sqrt{\left\|\sum_{s=1}^{t-k}(1-\beta)^{t-s}G_s\right\|^2 + \sum_{s=1}^{k}2(1-\beta)^{2s-2}\eta^2 L_7^2 d}\right]$$

$$+ \sqrt{2d}\eta L_8 \sum_{s=t+1-k}^{t}(1-\beta)^{t-s}\mathbb{E}\left[\|\nabla f(\mathbf{x}_s)\|\right]$$

$$\leq \mathbb{E}\left[\mathbb{E}_{i_{t-k}}\left[\sqrt{\left\|\sum_{s=1}^{t-k}(1-\beta)^{t-s}G_s\right\|^2 + \sum_{s=1}^{k}2(1-\beta)^{2s-2}\eta^2 L_7^2 d}\right]\right]$$

$$+ \sqrt{2d}\eta L_8 \sum_{s=t+1-k}^{t}(1-\beta)^{t-s}\mathbb{E}\left[\|\nabla f(\mathbf{x}_s)\|\right]$$

$$= \mathbb{E}\left[\mathbb{E}_{i_{t-k}}\left[\sqrt{\left\|\sum_{s=1}^{t-k-1}(1-\beta)^{t-s}G_s + (1-\beta)^k G_{t-k}\right\|^2 + \sum_{s=1}^{k}2(1-\beta)^{2s-2}\eta^2 L_7^2 d}\right]\right]$$

$$+ \sqrt{2d}\eta L_8 \sum_{s=t+1-k}^{t}(1-\beta)^{t-s}\mathbb{E}\left[\|\nabla f(\mathbf{x}_s)\|\right]$$

$$\leq \mathbb{E}\left[\sqrt{\mathbb{E}_{i_{t-k}}\left\|\sum_{s=1}^{t-k-1}(1-\beta)^{t-s}G_s + (1-\beta)^k G_{t-k}\right\|^2 + \sum_{s=1}^{k}2(1-\beta)^{2s-2}\eta^2 L_7^2 d}\right]$$

$$+ \sqrt{2d}\eta L_8 \sum_{s=t+1-k}^{t}(1-\beta)^{t-s}\mathbb{E}\left[\|\nabla f(\mathbf{x}_s)\|\right]$$

$$= \mathbb{E}\left[\sqrt{\left\|\sum_{s=1}^{t-k-1}(1-\beta)^{t-s}G_s\right\|^2 + (1-\beta)^{2k}\mathbb{E}_{i_{t-k}}\left[\|G_{t-k}\|^2\right] + \sum_{s=1}^{k}2(1-\beta)^{2s-2}\eta^2 L_7^2 d}\right]$$

$$+ \sqrt{2d}\eta L_8 \sum_{s=t+1-k}^{t}(1-\beta)^{t-s}\mathbb{E}\left[\|\nabla f(\mathbf{x}_s)\|\right]$$

$$\leq \mathbb{E}\left[\sqrt{\left\|\sum_{s=1}^{t-k-1}(1-\beta)^{t-s}G_s\right\|^2 + 2(1-\beta)^{2k}\eta^2 L_7^2 d + \sum_{s=1}^{k}2(1-\beta)^{2s-2}\eta^2 L_7^2 d}\right]$$

$$+ \sqrt{2d}\eta L_8 (1-\beta)^k \mathbb{E}\left[\|\nabla f(\mathbf{x}_{t-k})\|\right] + \sqrt{2d}\eta L_8 \sum_{s=t+1-k}^{t}(1-\beta)^{t-s}\mathbb{E}\left[\|\nabla f(\mathbf{x}_s)\|\right]$$

$$= \mathbb{E}\left[\sqrt{\left\|\sum_{s=1}^{t-k-1}(1-\beta)^{t-s}G_s\right\|^2 + \sum_{s=1}^{k+1}2(1-\beta)^{2s-2}\eta^2 L_7^2 d}\right]$$

$$+ \sqrt{2d}\eta L_8 \sum_{s=t-k}^{t}(1-\beta)^{t-s}\mathbb{E}\left[\|\nabla f(\mathbf{x}_s)\|\right],$$

where the third inequality is due to the following:

$$\mathbb{E}_{i_{t-k}} \left[ \|G_{t-k}\|^2 \right]$$

$$= \mathbb{E}_{i_{t-k}} \left[ \left\| \nabla f_{i_{t-k}}(\mathbf{x}_{t-k}) - \nabla f_{i_{t-k}}(\mathbf{x}_{t-k-1}) + \nabla f(\mathbf{x}_{t-k-1}) - \nabla f(\mathbf{x}_{t-k}) \right\|^2 \right]$$

$$= \mathbb{E}_{i_{t-k}} \left[ \left\| \nabla f_{i_{t-k}}(\mathbf{x}_{t-k}) - \nabla f_{i_{t-k}}(\mathbf{x}_{t-k-1}) \right\|^2 \right] + \left[ \left\| \nabla f(\mathbf{x}_{t-k-1}) - \nabla f(\mathbf{x}_{t-k}) \right\|^2 \right]$$

$$\quad - 2\mathbb{E}_{i_{t-k}} \left[ \left\langle \nabla f_{i_{t-k}}(\mathbf{x}_{t-k}) - \nabla f_{i_{t-k}}(\mathbf{x}_{t-k-1}), \nabla f(\mathbf{x}_{t-k}) - \nabla f(\mathbf{x}_{t-k-1}) \right\rangle \right]$$

$$= \mathbb{E}_{i_{t-k}} \left[ \left\| \nabla f_{i_{t-k}}(\mathbf{x}_{t-k}) - \nabla f_{i_{t-k}}(\mathbf{x}_{t-k-1}) \right\|^2 \right] - \left[ \left\| \nabla f(\mathbf{x}_{t-k-1}) - \nabla f(\mathbf{x}_{t-k}) \right\|^2 \right]$$

$$\leq \mathbb{E}_{i_{t-k}} \left[ \left\| \nabla f_{i_{t-k}}(\mathbf{x}_{t-k}) - \nabla f_{i_{t-k}}(\mathbf{x}_{t-k-1}) \right\|^2 \right]$$

For $\alpha \in (0,1)$, denoting $L_7^2 = (K_0 + K_1 + K_2)^2$, and $L_8^2 = K_1^2$, we have:

$$\mathbb{E}_{i_{t-k}} \left[ \left\| \nabla f_{i_{t-k}}(\mathbf{x}_{t-k}) - \nabla f_{i_{t-k}}(\mathbf{x}_{t-k-1}) \right\|^2 \right]$$

$$\leq \|\mathbf{x}_{t-k} - \mathbf{x}_{t-k-1}\|^2 \left( K_0 + K_1 \|\nabla f(\mathbf{x}_{t-k})\|^\alpha + K_2 \|\mathbf{x}_{t-k} - \mathbf{x}_{t-k-1}\|^{\frac{\alpha}{1-\alpha}} \right)^2$$

$$\leq \eta^2 d \left( K_0 + K_1 + K_2 + K_1 \|\nabla f(\mathbf{x}_{t-k})\| \right)^2$$

$$\leq 2\eta^2 d \left( K_0 + K_1 + K_2 \right)^2 + 2\eta^2 d K_1^2 \|\nabla f(\mathbf{x}_{t-k})\|^2$$

$$\leq 2\eta^2 d L_7^2 + 2\eta^2 d L_8^2 \|\nabla f(\mathbf{x}_{t-k})\|^2,$$

where the second inequality holds by setting $\eta \leq d^{-1/2}$ such that $\|\mathbf{x}_{t-k} - \mathbf{x}_{t-k-1}\| \leq \eta\sqrt{d} \leq 1$.
For $\alpha = 1$, denoting $L_7^2 = 9L_0^2$ and $L_4^2 = 9L_1^2$, we have:

$$\mathbb{E}_{i_{t-k}} \left[ \left\| \nabla f_{i_{t-k}}(\mathbf{x}_{t-k}) - \nabla f_{i_{t-k}}(\mathbf{x}_{t-k-1}) \right\|^2 \right]$$

$$\leq 2 \|\mathbf{x}_{t-k} - \mathbf{x}_{t-k-1}\|^2 \left( L_0^2 + L_1^2 \|\nabla f(\mathbf{x}_{t-k})\|^2 \right) \left( \exp \left( L_1^2 \|\mathbf{x}_{t-k} - \mathbf{x}_{t-k-1}\|^2 \right) \right)^2$$

$$\leq 18\eta^2 d \left( L_0^2 + 2L_1^2 \|\nabla f(\mathbf{x}_{t-k})\|^2 \right)$$

$$\leq 2\eta^2 d L_7^2 + 2\eta^2 d L_8^2 \|\nabla f(\mathbf{x}_{t-k})\|^2,$$

where the second inequality holds by setting $\eta \leq \frac{1}{\sqrt{L_1^2 d}}$, such that we have $\|\mathbf{x}_{t-k} - \mathbf{x}_{t-k-1}\|^2 \leq \eta^2 d \leq \frac{1}{L_1^2}$.

## F   Additional experiments

In this section, we present additional experiments on the CIFAR-10 dataset to validate whether the proposed SSVR method is sensitive to hyper-parameters such as learning rate $\eta$, momentum parameter $\beta$, and batch size. Specifically, we fix the value of $\beta$ as 0.5 and try different learning rates from the set $\{5e-3, 1e-3, 5e-4, 1e-4, 5e-5\}$. Then, we fix the learning rate as $1e-3$ and enumerate $\beta$ from the set $\{0.3, 0.5, 0.7, 0.9, 0.99\}$. The results are reported in Figure 3 and Figure 4, respectively, which indicate that our method is insensitive to the choice of hyper-parameters within a certain range. Finally, we also try different batch sizes from the set $\{64, 128, 256, 512\}$, and the results are shown in Figure 5. It can be seen that our algorithm does not necessitate large batches for convergence and is not sensitive to variations in batch sizes.

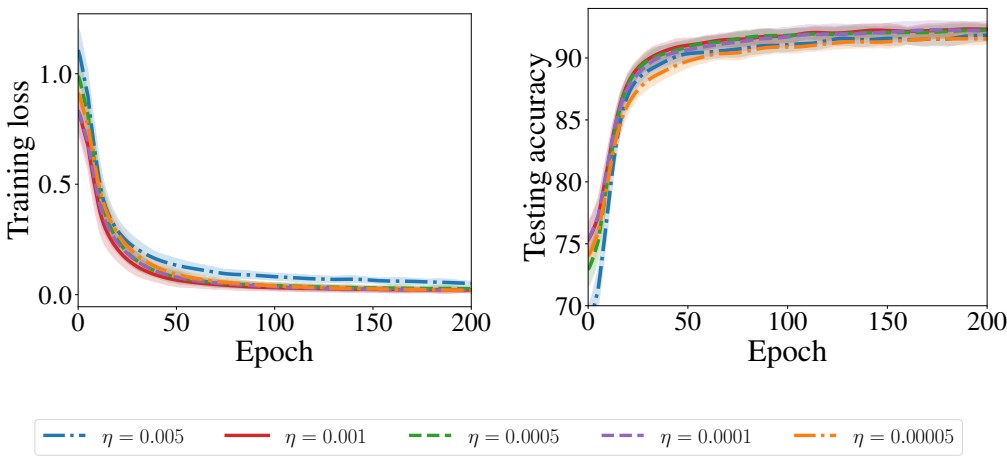

Figure 3: Results for CIFAR-10 dataset with different learning rates.

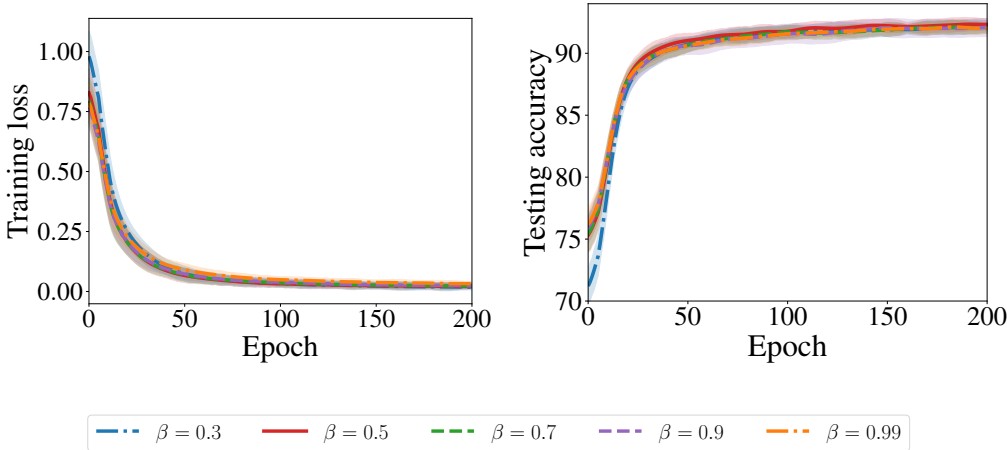

Figure 4: Results for CIFAR-10 dataset with different $\beta$.

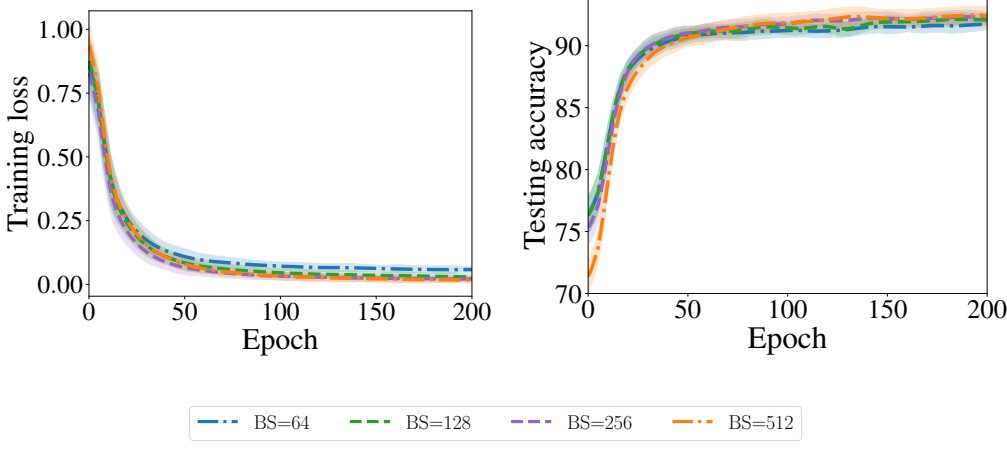

Figure 5: Results for CIFAR-10 dataset with different batch sizes.

