# OpenReview forum: "Efficient Sign-Based Optimization: Accelerating Convergence via Variance Reduction"
_NeurIPS.cc/2024/Conference — NeurIPS 2024 poster_

### Official Review · Reviewer_XovB · 2024-06-28

**Soundness:** 3
**Presentation:** 3
**Contribution:** 3
**Rating:** 7
**Confidence:** 5

**Summary:**

This paper introduces the Sign-based Stochastic Variance Reduction (SSVR) algorithm, which enhances the convergence rate of the traditional signSGD method. By incorporating variance reduction techniques with sign-based updates, the authors achieve a convergence rate of $O(d^{1/2}T^{-1/3})$ for general stochastic optimization and $O(m^{1/4}d^{1/2}T^{-1/2})$ for finite-sum problems. Additionally, the paper proposes novel algorithms for distributed environments by introducing the unbiased sign operation, resulting in superior convergence rates for heterogeneous data. Numerical experiments further validate the effectiveness of the proposed methods.

**Strengths:**

1. The proposed methods improve the convergence rates over traditional signSGD methods and their variants, achieving faster convergence rates for general non-convex optimization and finite-sum optimization.
2. The SSVR-MV algorithm developed in the paper is communication-efficient and well-suited for distributed settings. The obtained convergence rates significantly enhance previous results in heterogeneous settings.
3. Numerical experiments validate the effectiveness of the proposed methods, demonstrating superior performance compared to existing sign-based optimization methods in terms of convergence speed and accuracy.

**Weaknesses:**

1. Given that signSGD methods typically require large batch sizes to ensure convergence, the authors should include more experimental results to demonstrate the dependency on batch size for the proposed methods. This would clarify whether the proposed method also necessitates large batches for convergence in practice.
2. The authors introduce the stochastic unbiased sign operation in Definition 1, which differs from the traditional sign operation. It would be beneficial to provide a more detailed explanation of their differences and specify when the sign operation is preferred.
3. There are some typos in the paper, as listed below:
   - Page 2, Lines 44 and 45: $T^{1/2}$ should be $T^{-1/2}$.
   - Page 3, Line 98: "signed-based" should be "sign-based".
   - Page 5, Algorithm 2, Step 4: "set $t=\tau$" should be "set $\tau=t$".
   - Page 9, Figure 2: "SSVR" should be "SSVR-MV".
   - Page 28, Line 482: $f(x_t;\xi_t^j)$ should be $\nabla f(x_t;\xi_t^j)$.

**Questions:**

1. Can you provide a more detailed discussion about the design of equation (3), especially the error correction term? The authors claim that "This gap can be effectively mitigated by the error correction term we introduced in equation (3)," but it remains unclear how it works based solely on the content in the main body.
2. Could you provide additional experimental results to demonstrate the performance of the proposed SSVR method with different batch sizes?

**Limitations:**

The paper is theoretical and does not present potential negative societal impacts.

---

> ### Author Rebuttal · Authors · 2024-08-05
>
> Thank you very much for your constructive comments!
>
> ---
> **Q1:** The authors should include more experimental results to demonstrate the dependency on batch size for the proposed methods. This would clarify whether the proposed method also necessitates large batches for convergence in practice.
>
> **A1:** According to your request, we have included additional experimental results for different batch sizes {$64,128,256,512$}, and the results can be found in the **Global Response**. As can be seen, the proposed method does not necessitate large batches for convergence in practice.
>
> ---
>
> **Q2:** The authors introduce the stochastic unbiased sign operation in Definition 1, which differs from the traditional sign operation. It would be beneficial to provide a more detailed explanation of their differences and specify when the sign operation is preferred.
>
> **A2:** The main difference is that the traditional sign operation produces a biased estimation of the original input, whereas the stochastic sign operation in Definition 1 provides an unbiased estimation. The disadvantage of the unbiased sign operation is that it requires the bounded gradient assumption to ensure the input of this operation is bounded. In the centralized setting, where the traditional sign operation already achieves improved convergence rates, we simply use it to avoid introducing additional assumptions. In heterogeneous distributed settings, where we need to use the sign operation twice, the traditional sign operation  can lead to large bias and worse convergence rates. Therefore, we use the stochastic sign operation to obtain better convergence rates in this setting.
>
> ---
>
> **Q3:** There are some typos in the paper.
>
> **A3:** Thank you for catching typos. We will correct them in the revised version.
>
> ---
>
> **Q4:** Can you provide a more detailed discussion about the design of equation (3), especially the error correction term? The authors claim that "This gap can be effectively mitigated by the error correction term we introduced in equation (3)," but it remains unclear how it works based solely on the content in the main body.
>
> **A4:** According to the analysis from Lines 379 to 380, we can bound the gradient estimation error as follows:
> \begin{align}
>         \mathbb{E}  \left[ || \nabla f(\mathbf{x}_{t+1}) - \mathbf{v}\_{t+1} ||^2\right] \leq  (1-\beta) \mathbb{E}\left[ ||\mathbf{v}_t - \nabla f(\mathbf{x}_t) ||^2\right]   + 2L^2 \mathbb{E}\left[||\mathbf{x}\_{t+1} - \mathbf{x}_t ||^2\right]  + 2\beta^2 \mathbb{E} \left[||\nabla f\_{i\_{t+1}}  ( \mathbf{x}\_{t+1} )  - \nabla f ( \mathbf{x}\_\{t+1} ) + \nabla f( \mathbf{x}\_{\tau} ) - \nabla f\_{i\_{t+1}} ( \mathbf{x}\_{\tau} ) ||^2\right]
> \end{align}
> Note that in the last term, $ \nabla f(\mathbf{x}\_{\tau})- \nabla f\_{i\_{t+1}}(\mathbf{x}\_{\tau})$ is the error correction term. Without this term, we would need to bound $ \mathbb{E} \left[||\nabla f\_{i\_{t+1}}(\mathbf{x}\_{t+1}) - \nabla f(\mathbf{x}\_{t+1})||^2\right]$, which is hard to deal with without assuming $ \mathbb{E} \left[||\nabla f\_{i}(\mathbf{x}) - \nabla f(\mathbf{x})||^2\right] \leq \sigma^2$ for each function $f_i$. However, with the error correction term, we can effectively bound it as:
> \begin{align}
>      \mathbb{E} \left[||\nabla f\_{i\_{t+1}}(\mathbf{x}\_{t+1}) - \nabla f\_{i\_{t+1}}(\mathbf{x}\_{\tau})  - \nabla f(\mathbf{x}\_{t+1}) + \nabla f(\mathbf{x}\_{\tau})||^2\right]
>     \leq  \mathbb{E}\left[||\nabla f\_{i\_{t+1}}(\mathbf{x}\_{t+1}) - \nabla f\_{i\_{t+1}}(\mathbf{x}\_{\tau})||^2 \right]
>      \leq L^2 \mathbb{E}\left[|| \mathbf{x}\_{t+1} - \mathbf{x}\_{\tau}||^2 \right].
> \end{align}
> As long as the learning rate is small enough and $\tau$ is updated periodically, we can easily bound the above term.

---

> > ### Comment · Reviewer_XovB · 2024-08-13
> > **final rating**
> >
> > Thank you for the detailed responses. They have satisfactorily addressed all my concerns, and I am inclined to recommend acceptance.

---

### Official Review · Reviewer_Huhp · 2024-07-11

**Soundness:** 4
**Presentation:** 4
**Contribution:** 4
**Rating:** 7
**Confidence:** 4

**Summary:**

The paper introduces an enhanced sign-based method called SSVR, designed to improve the convergence rate of signSGD with variance reduction techniques. SSVR achieves an improved convergence rate of $O(d^{1/2}T^{-1/3})$ for stochastic non-convex functions. For non-convex finite-sum optimization problems, the SSVR-FS method demonstrates an improved convergence rate of $ O(m^{1/4}d^{1/2}T^{-1/2}) $. To address heterogeneous data in distributed settings, the authors present novel algorithms that outperform existing methods in terms of convergence rates. The effectiveness of these proposed methods is validated through numerical experiments.

**Strengths:**

1. The proposed methods improves upon existing algorithms in complexities. The sign-based algorithms have broad applications in the ML community and may be of great interest in the field.
2. The proposed sign-based variance-reduced estimator and stochastic unbiased sign operation are novel and efficient. They effectively reduce the gradient estimation error and are communication efficient, particularly for heterogeneous data in distributed settings.
3. The experimental results on various datasets validate the effectiveness of the proposed methods.
4. The paper is well-written, with clear problem settings and contributions. The proofs are also easy to follow.

**Weaknesses:**

1. The authors introduce the bounded gradient assumption (Assumption 4) for distributed settings. Why is this assumption necessary? Is it commonly used in previous literature for this setting?
2. The description of Algorithm 2 is not very rigorous. The variable $\tau$ in step 4 and $\tau$ in step 10 seems different but used the same notation.
3. Some typos:
    - Line 98: "signed-based" --> "sign-based."
    - Line 247: "Assumption 5′" --> "Assumption 5" or "Assumption 4′."
    - Line 487: "$S(v_t^j)$" --> "$S_{R}(v_t^j)$."

**Questions:**

1. Does previous work also require similar bounded gradient assumptions for the majority vote in heterogeneous settings?
2. Is the performance of the proposed method sensitive to variations in batch size?

**Limitations:**

This paper does not present negative societal impacts.

---

> ### Author Rebuttal · Authors · 2024-08-05
>
> Thank you very much for your constructive comments and suggestions!
>
> ---
>
> **Q1:** The authors introduce the bounded gradient assumption (Assumption 4) for distributed settings. Why is this assumption necessary? Is it commonly used in previous literature for this setting?
>
> **A1:** This assumption is necessary because the unbiased sign operation $\operatorname{S_R}(\cdot)$ defined in Definition 1 requires its input to be bounded such that $ ||\mathbf{v}||_{\infty} \leq R $. To meet this requirement, we assume the gradient is bounded so that the norm of our gradient estimator $\mathbf{v}_t^j$ is also bounded. Similar bounded (stochastic) gradient assumptions are also used in previous literature [Jin et al., 2023, Sun et al., 2023] for this heterogeneous distributed setting.
>
> ---
>
> **Q2:** The description of Algorithm 2 is not very rigorous. The variable $\tau$ in step 4 and $\tau$ in step 10 seems different but used the same notation.
>
> **A2:** Sorry for the misleading notation. We will clarify this by replacing the $\tau$ in step 10 with $\varphi$ in the revised version.
>
> ---
>
> **Q3:** Some typos.
>
> **A3:** Thank you for pointing out the typos. We will correct them in the revised paper.
>
> ---
>
>
> **Q4:** Does previous work also require similar bounded gradient assumptions for the majority vote in heterogeneous settings?
>
> **A4:** Yes, as mentioned in A1, previous literature [Jin et al., 2023, Sun et al., 2023] also requires similar bounded gradient assumptions for this setting.
>
> ---
>
> **Q5:** Is the performance of the proposed method sensitive to variations in batch size?
>
> **A5:** To answer this question, we conduct experiments with different batch sizes {$64, 128, 256, 512$}, and the results can be found in **Global Response**. As can be seen, the proposed algorithm is not sensitive to variations in batch size.
>
> ---
>
> **References:**
>
> Jin et al.  Stochastic-Sign SGD for federated learning with theoretical guarantees. 2023.
>
> Sun et al.  Momentum ensures convergence of SIGNSGD under weaker assumptions. ICML, 2023.

---

> > ### Comment · Reviewer_Huhp · 2024-08-09
> >
> > Thank the authors for the rely. It addresses all my questions and concerns. I'm willing to keep my score.

---

### Official Review · Reviewer_2Gd9 · 2024-07-11

**Soundness:** 3
**Presentation:** 3
**Contribution:** 3
**Rating:** 6
**Confidence:** 4

**Summary:**

This paper considers application of variance reduction to sign-based optimization, i.e., a setting where the algorithm can only access to $\mathrm{sign}(\nabla f(x^t; \xi^t))$ at step $t$, in the smooth and nonconvex setting.
Because $\\{\pm 1\\}$-valued vectors can be transmitted more efficiently than real-valued vectors, the sign-based optimization is considered to be useful especially in distributed optimization.
While a naive SGD-type algorithm previously achieved the convergence rate of $O(d^\frac12 T^{-\frac14})$, the proposed variance-reduced algorithm yields the rate of $O(d^\frac12 T^{-\frac13})$ (measured by the $L_1$-norm of the gradient.)
Next, they applied the algorithm to finite-sum optimization (with $m$ components) and obtained the convergence rate of $O(m^\frac14 d^\frac12 T^{-\frac14})$.
Finally, they applied these results to distributed optimization (the server received $\\{\pm 1\\}$-valued gradient estimators from clients), with an additional technique called \textit{majority vote} to obtain unbiased gradient estimators, and obtained improved communication complexities.

**Strengths:**

### **The problem is well-motivated**

Sign-based optimization is an important technique to reduce computational / communication complexities in large-scale optimization. I think it is a natural attempt to apply variance reduction, which is a common technique to accelerate optimization, to achieve more efficient sign-based optimization.

### **Improved convergence rate**

Overall, the proposed approach improved the convergence rates of sign-based optimization, which I think solid contributions to the theory of optimization.

- For stochastic non-convex optimization, the proposed algorithm achieves $\min{1\leq t \leq T}\mathbb{E}[\\|\nabla f(x^t)\\|]\lesssim d^\frac12 T^{-\frac13}$, which is an improvement from the $d^\frac12 T^{-\frac14}$ rate of SignSGD ([Bernstein et al., 2018](https://arxiv.org/abs/1802.04434)).

- For finite-sum optimization, the proposed algorithm achieves $\min{1\leq t \leq T}\mathbb{E}\[\|\nabla f(x^t)\\|]\lesssim m^\frac14 d^\frac12 T^{-\frac12}$, whereas the previous SignSVRG algorithm achieves the rate of $O(m^\frac12 d^\frac12 T^{-\frac12})$ ([Chzhen and Schechtman, 2023](https://arxiv.org/abs/2305.13187). (I am a bit confused because Ii looks like there is no benefit of variance reduction for SignSVRG in terms of the dependency on $m$.)

- For distributed optimization, the proposed algorithm achieves $\min{1\leq t \leq T}\mathbb{E}[\\|\nabla f(x^t)\\|]\lesssim d^\frac12 T^{-\frac12} + d n^{-\frac12}$ or $\lesssim d^\frac14 T^{-\frac14}$. The previous bound was $d T^{-\frac14} + d n^{-\frac12}$ ([Sun et al., 2023](https://proceedings.mlr.press/v202/sun23l.html)) or $d^\frac38 T^{-\frac18}$ ([Jin et al., 2023](https://arxiv.org/abs/2002.10940), in $L^2$-norm). This is achieved by a trick called majority vote, which stochastically maps the real-valued gradient vector (or $\nabla f_i(x)$) into $\\{\pm 1\\}$-valued vectors transmitted to the server.

### **Experimental results**

The authors showed sufficient amount of experiments to verify the validity of the proposed methods (as a theory paper).

**Weaknesses:**

### **Technical novelty**

Variance reduction itself is quite common in the optimization field so I am afraid that this paper might be a bit incremental. Section 3.3 (Results under weaker assumptions) looks redundant given Theorem 2, as there are no technical difficulty about weakening the assumption described.

### **Hyperparameter choice**

Although this is a common criticism, most variance reduction algorithm require a very careful hyperparameter tuning. It looks like the theoretical guarantees of the proposed algorithms also require a specific set of hyperparameters, and I am not confident in the robustness of the proposed algorithms to the hyperparameters.

**Questions:**

### **On additional terms**

- Why is the term of $md/T$ required in Theorem 4?

- Why is the term of $d n^{-\frac12}$ required in Theorem 5?

### **On the difficulties to further improve the convergence rates**

- In theorem 6, why cannot you improve the convergence rate into $\mathrm{poly}(d) T^{-\frac13}$?

### **Assumptions**

- Is it possible to weaken Assumption 3 to averaged smoothness, i.e., $\frac1m \sum_{i=1}^m \\|\nabla f_i(x)-\nabla f_i(y)\\|^2 \leq L^2 \\|x-y\\|^2$

---

> ### Author Rebuttal · Authors · 2024-08-05
>
> Thanks for your constructive comments! We will revise accordingly.
>
> ---
> **Q1:** Variance reduction is quite common in the optimization field so I am afraid this paper might be a bit incremental. Section 3.3 looks redundant given Theorem 2, as there are no technical difficulty about weakening the assumption described.
>
> **A1:** First, we want to emphasize that our Algorithm 1 is the first sign-based variance reduction method to improve the convergence rate from $\mathcal{O}(T^{-1/4})$ to $\mathcal{O}(T^{-1/3})$ (Theorem 1). Second, to deal with the finite-sum problem, we introduce a novel error correction term into our gradient estimator in Algorithm 2, making our approach distinct from existing methods. Furthermore, the corresponding guarantee (Theorem 2) surpasses the existing variance reduction method signSVRG. Finally, the proposed SSVR-MV algorithm and its corresponding analysis (Theorem 3 & 4) are also new, where we utilize unbiased sign operations and the projection operation in the SSVR-MV method. As a result, we believe our paper provides significant contributions beyond being merely incremental in the existing literature.
>
> Regarding Section 3.3, we agree it is not the main contribution of our paper and our intention is to broaden the applicability of our methods to a wider range of functions by relaxing assumptions. Moreover, there actually exists technique challenges in the analysis. For instance, in the previous analysis, we can bound $\mathbb{E}[||\nabla f(\mathbf{x}\_{t+1};\xi\_{t+1})-\nabla f(\mathbf{x}_t;\xi\_{t+1})||^2]\le L^2||\mathbf{x}\_{t+1}-\mathbf{x}_t||^2\le L^2 \eta^2d$ with small $\eta$. Under weaker assumptions, we can only ensure $\mathbb{E}[||\nabla f(\mathbf{x}\_{t+1};\xi\_{t+1})-\nabla f(\mathbf{x}_t;\xi\_{t+1})||^2]\le ||\mathbf{x}\_{t+1}-\mathbf{x}_t ||^2(K_0+K_1\mathbb{E}\_{\xi\_{t+1}}||\nabla f(\mathbf{x}_t;\xi\_{t+1})||)$, which can not be bounded since $||\nabla f(\mathbf{x}_t;\xi\_{t+1})||$ is unbounded. To address this, we employed a different analysis, using mathematical induction techniques (see the proofs of Lemmas 3 and 4). Therefore, the analyses for Section 3.3 are more challenging. However, following your suggestion, we will consider moving Section 3.3 to the appendix.
>
> ---
> **Q2:** I am not confident in the robustness of the proposed algorithms to hyper-parameters.
>
> **A2:** Following your suggestion, we have examined the behavior of our algorithm with different values of the learning rate and parameter $\beta$. Initially, we fix $\beta=0.5$ and vary the learning rate within the set {$5e-3,1e-3,5e-4,1e-4,5e-5$}. Then, we fix the learning rate at $1e-3$ and varied $\beta$ within {$0.3,0.5,0.7,0.9,0.99$}. The results, which are reported in the **Global Response**, indicate that our method is not sensitive to the choice of hyper-parameters.
>
> ---
> **Q3:** Why is the term of $md/T$ required in Theorem 4?
>
> **A3:** Compared to Assumption 3 used in Theorem 2, we use the weaker Assumption 3$^\prime$ in Theorem 4, which includes an additional $||\nabla f(\mathbf{x}_t)||$ term. Consequently, this extra term appears in the analysis. In the final step, we have $\mathbb{E}[\frac{1}{T}\sum\_{t=1}^T||f(\mathbf{x}_t)||_1]\le\mathcal{O}(\frac{1}{\eta T}+\eta d+d\sqrt{m}\eta)+\mathcal{O}(d{\eta m})\mathbb{E}[\frac{1}{T}\sum\_{t=1}^T||\nabla f(\mathbf{x}_t)||_1]$. To cancel the last term, we set $\eta\leq\mathcal{O}(\frac{1}{md})$. As a result, we have $\mathcal{O}(\frac{1}{\eta T})=\mathcal{O}(\frac{md}{T})$ in the convergence rate.
>
> ---
> **Q4:** Why is the term of $dn^{-1/2}$ required in Theorem 5?
>
> **A4:** According to equation (15) on Page 29, we need to bound $R\sqrt{d}||\frac{\nabla f(\mathbf{x}_t)}{R}-\frac1n\sum\_{j=1}^n S(\mathbf{v}_t^j)||$. To this end, we separate it into two terms: $R\sqrt{d}||\frac{\nabla f(\mathbf{x}_t)}{R}-\frac{1}{nR}\sum\_{j=1}^n\mathbf{v}_t^j||$ and $ R\sqrt{d}||\frac{1}{n}\sum\_{j=1}^n(S(\mathbf{v}_t^j)-\frac{\mathbf{v}_t^j}{R})||$. The first term represents the estimation error of estimator $\mathbf{v}_t^j$ and can be bounded using a similar technique in Theorem 1. The second term leads to the $\mathcal{O}(dn^{-1/2})$ term, since we bound it as:
> \begin{align} R\sqrt{d}\mathbb{E}[||\frac1n\sum\_{j=1}^n (S(\mathbf{v}_t^j)-\frac{\mathbf{v}_t^j}{R})||]\le R\sqrt{d}\sqrt{\mathbb{E}[||\frac{1}{n}\sum\_{j=1}^n(S(\mathbf{v}_t^j)-\frac{\mathbf{v}_t^j}{R})||^2]}\le R\sqrt{d}\sqrt{\frac{1}{n^2}\sum\_{j=1}^n\mathbb{E}||S(\mathbf{v}_t^j)-\frac{\mathbf{v}_t^j}{R}||^2}\le R\sqrt{d}\sqrt{\frac{1}{n^2}\sum\_{j=1}^n\mathbb{E}[||S(\mathbf{v}_t^j)||^2]}\le \frac{dR}{\sqrt{n}}.\end{align}
>
> ---
> **Q5:** In theorem 6, why cannot you improve the convergence rate to $\mathcal{O}(T^{-1/3})$?
>
> **A5:** For Theorem 1, we obtain the convergence rate of $\mathcal{O}(T^{-1/3})$ with the help of Assumptions 1 and 2. In theorem 6, although the proposed unbiased sign operation provides an unbiased gradient estimation, the obtained gradient cannot satisfy the average smoothness (Assumption 1), which is crucial for achieving the $\mathcal{O}(T^{-1/3})$ converge rate. This is the main reason why we cannot further improve the convergence rate.
>
> ---
> **Q6:** Is it possible to weaken Assumption 3 to averaged smoothness, i.e., $\frac1m \sum_{i=1}^m||\nabla f_i(x)-\nabla f_i(y)||^2\le L^2||x-y||^2$?
>
> **A6:** Yes, it is possible. In the analysis, Assumption 3 is used to ensure $\mathbb{E}\_{i\_{t+1}}[||\nabla f\_{i\_{t+1}}(\mathbf{x}\_{t+1})-\nabla f\_{i\_{t+1}}(\mathbf{x}\_t)||^2]\le L^2||\mathbf{x}\_{t+1}-\mathbf{x}\_t||^2$. With the averaged smoothness, we can achieve a similar guarantee. Since $i_{t+1}$ is randomly sampled from {$1,\cdots,m$}, we have:
> \begin{align}\mathbb{E}\_{i\_{t+1}}[||\nabla f\_{i\_{t+1}}(\mathbf{x}\_{t+1})-\nabla f\_{i\_{t+1}}(\mathbf{x}\_t)||^2]=\frac1m\sum_{i=1}^{m}||\nabla f_i(\mathbf{x}\_{t+1})-\nabla f_{i}(\mathbf{x}\_t)||^2\le L^2 ||\mathbf{x}\_{t+1}-\mathbf{x}\_t||^2,\end{align}
> which indicates the proof holds with averaged smoothness.

---

> > ### Comment · Reviewer_2Gd9 · 2024-08-12
> >
> > Thank you very much for your detailed response. I would like to keep my score.
> >
> > Best,
> > reviewer.

---

### Official Review · Reviewer_hYmq · 2024-07-16

**Soundness:** 3
**Presentation:** 3
**Contribution:** 3
**Rating:** 7
**Confidence:** 3

**Summary:**

This paper investigates the stochastic optimization problem and a special case of finite sum optimization problem. They propose the
Sign-based Stochastic Variance Reduction (SSVR) method, leveraging variance reduction techiniques, which improves the convergence rate of Sign stochastic gradient descent (signSGD) algorithm.
Further, they also study the heterogeneous majority vote in distributed settings and introduce two novel algorithms, whose convergence rates improve the state-of-the-art.
Numerical experiments prove the efficiency of their algorithms and support their theoretical guarantees.

**Strengths:**

The paper is clearly written, with detailed discussion and comparision.

**Weaknesses:**

Some places of the paper need further clarification.

**Questions:**

1. How did you get the convergence rate of $O(m^{1/2}d^{1/2}T^{−1/2})$ for the SignSVRG of y? It seems that in their paper they claim $O(T^{-1/2}) convergence rate without $m$ and $n$ in the numerator. Did you calculate this result by yourself? Please provide discussions.

2. There is another paper studying sign-based optimization (Qin et al. (2023)), the setting studied in this paper is the same as your problem (2).  Can you also discuss this paper and compare the results?


References:
1. E. Chzhen and S. Schechtman. SignSVRG: fixing SignSGD via variance reduction. ArXiv e-prints, 317 arXiv:2305.13187, 2023.
2. Qin, Z., Liu, Z., & Xu, P. (2023). Convergence of Sign-based Random Reshuffling Algorithms for Nonconvex Optimization. arXiv preprint arXiv:2310.15976.

**Limitations:**

Yes.

---

> ### Author Rebuttal · Authors · 2024-08-05
>
> Many thanks for the constructive feedback! We will revise our paper accordingly.
>
> ---
>
> **Q1:** How did you get the convergence rate of $\mathcal{O}(m^{1/2}d^{1/2}T^{-1/2})$ for the SignSVRG? It seems that in their paper they claim $\mathcal{O}(T^{-1/2})$ convergence rate without $m$ and $d$ in the numerator. Did you calculate this result by yourself? Please provide discussions.
>
> **A1:** Yes, the original paper does not explicitly state the dependence on $m$ and $d$, so we derived these dependencies ourselves. In Remark 2 of their paper, they indicate that $\mathbb{E}\left[||\nabla f(\bar{x}\_{T^{\prime}})||_{1}\right]  \lesssim  \sqrt{\frac{d}{T^{\prime}}}  \max \left( ( f\left(x_1\right)-f\left(x\_{*}\right)+1 ) / d, \mathrm{P} \right)$, where $T^{\prime}$ is the iteration number. Consequently, to ensure $\mathbb{E} \left[ || \nabla f(\bar{x}\_{T^{\prime}})||\_{1} \right] \leq \epsilon$, we require at least $T^{\prime}=\mathcal{O}\left( dP^2 \epsilon^{-2}\right)$. As mentioned in Lemma 4 of their paper, the algorithm may need to recompute the full gradient ($m$ samples) every $P$ round, resulting in a sample complexity on the order of $\mathcal{O}\left( d m \epsilon^{-2}\right)$. This leads to the overall convergence rate of $\mathcal{O}\left( d^{1/2} m^{1/2} T^{-1/2}\right)$.
>
> ---
>
> **Q2:** There is another paper studying sign-based optimization [Qin et al, 2023], the setting studied in this paper is the same as your problem (2). Can you also discuss this paper and compare the results?
>
> **A2:** Thank you for bringing this related work to our attention. We will cite this paper and provide a detailed discussion in the revised version of our paper. Specifically, the mentioned paper [Qin et al., 2023] focuses on the finite-sum optimization problem and investigates signSGD **with random reshuffling**, achieving a convergence rate of $\mathcal{O}\left({\log (mT)}/{\sqrt{mT}}+||\sigma ||_1\right)$, where $m$ is the number of component functions, $T$ is the number of epochs of data passes, and $\sigma$ is the variance bound of stochastic gradients. By leveraging variance-reduced gradients and momentum updates, they further propose the SignRVR and SignRVM methods, both achieving the convergence rate of $\mathcal{O}\left({\log (mT)\sqrt{m}}/{\sqrt{T}}\right)$. In contrast, our method obtains a convergence rate of $\mathcal{O}\left({m^{1/4}}/{\sqrt{T}}\right)$, which enjoys a better dependence on $m$ than their methods.
>
> ---
>
> **References:**
>
> Qin et al.  Convergence of Sign-based Random Reshuffling Algorithms for Nonconvex Optimization. 2023.

---

> > ### Comment · Reviewer_hYmq · 2024-08-11
> >
> > Thanks for the rebuttal. It addressed all my concerns. I will increase my score.

---

> > > ### Author Response · Authors · 2024-08-11
> > >
> > > Thank you very much for your kind reply! We will revise our paper according to the constructive reviews.
> > >
> > > Best regards,
> > >
> > > Authors

---

### Author Rebuttal · Authors · 2024-08-05

## **Global Response** ##

---
In response to the request of reviewers, we provide additional experimental results in this part.

**Figure 1 \& Figure 2:** According to the suggestion of Reviewer 2Gd9, we provide the performance of our SSVR method with different hyper-parameters. First, we fix the value of $\beta$ as 0.5 and try different learning rates from the set {$5e-3, 1e-3, 5e-4, 1e-4, 5e-5$}. Then, we fix the learning rate as $1e-3$ and enumerate $\beta$ from the set {$0.3, 0.5,0.7,0.9,0.99$}. The results show that our method is insensitive to the choice of the hyper-parameters within a certain range.

**Figure 3:** As requested by Reviewer Huhp and Reviewer Xovb, we include experiments on the SSVR method with different batch sizes {$64, 128, 256, 512$}. The results indicate that our algorithm does not necessitate large batches for convergence and is not sensitive to variations in batch sizes.

---

### Comment · Area_Chair_aAQH · 2024-08-08
**Please read the authors’ rebuttal and reply by August 13, 2024 (11:59 PM, AOE time)**

Dear Reviewers,

Thank you for your hard work during the review process. The authors have responded to your initial reviews. **If you haven’t already done so, please take the time to carefully read their responses.** It is crucial for the authors to receive acknowledgment of your review by the deadline for the author-reviewer discussion period, which is August 13, 2024 (11:59 PM, Anywhere on Earth). Please address any points of disagreement with the authors as early as possible.

Best,

Your AC

---

### Decision · Program_Chairs · 2024-09-25

**Decision:**

Accept (poster)

**Comment:**

This paper investigates the convergence of sign-SGD algorithms, with a focus on enhancing their performance through variance reduction techniques. The authors successfully improve the convergence rate of sign-SGD for stochastic optimization from \(O(d^{1/2}T^{-1/4})\) to \(O(d^{1/2}T^{-1/3})\). For finite-sum problems, the proposed approach further accelerates convergence by a factor of \(m^{1/4}\), where \(m\) represents the number of individual functions. Additionally, the paper demonstrates that this technique can be effectively applied to distributed settings, improving upon existing results in the literature.

The paper is generally well-written, presenting a solid contribution in the form of improved convergence rates over established methods. Following an in-depth discussion between the authors and reviewers, consensus has been reached that the paper is suitable for publication. I am also recommending acceptance of this paper. However, I have some additional comments that I believe could enhance the paper’s clarity and presentation, which I encourage the authors to consider in their final version:

1. **Clarification of Convergence Rate Definitions:**
   The paper would benefit from a more detailed description of the convergence rate definitions used. Specifically, it would be helpful if the authors could clearly delineate the differences between this work and others in terms of convergence rate definitions—whether in terms of the norm of the gradient or the squared norm, and whether considering \(l_1\) or \(l_2\) norms. This clarification will aid readers in making direct comparisons between the results in this paper and those in related works.

2. **Discussion on Upper vs. Lower Bounds:**
   In the introduction, the authors discuss lower bounds for stochastic gradient methods and finite-sum optimization. However, the upper bounds provided in the theoretical section are primarily for the \(l_1\) norm, while the literature often discusses upper bounds in terms of the \(l_2\) norm. The authors should add a discussion that addresses this discrepancy, clarifying that it is acceptable that there is no exact lower bound for the sign-based optimization setting.

3. **Comprehensive Discussion of Variance Reduction Techniques:**
   Since SVRG plays a pivotal role in achieving the improved convergence results in this paper, the related work section would benefit from a more thorough discussion of key variance reduction algorithms such as SCSG [1], SDCA [2, 4], nonconvex SVRG [3], SNVRG [5], etc.

[1] Lei, Lihua, and Michael Jordan. "Less than a single pass: Stochastically controlled stochastic gradient." Artificial Intelligence and Statistics. PMLR, 2017.

[2] Shalev-Shwartz, Shai, and Tong Zhang. "Stochastic dual coordinate ascent methods for regularized loss minimization." Journal of Machine Learning Research 14.1 (2013).

[3] Reddi, Sashank J., et al. "Stochastic variance reduction for nonconvex optimization." International conference on machine learning. PMLR, 2016.

[4] Richtárik, Peter, and Martin Takáč. "Iteration complexity of randomized block-coordinate descent methods for minimizing a composite function." Mathematical Programming 144.1 (2014): 1-38.

[5] Zhou, Dongruo, et al. "Stochastic nested variance reduction for nonconvex optimization." Journal of machine learning research 21.103 (2020): 1-63.